# Injectable therapeutic system incorporating neurogenesis-programmed stem cells concomitantly promoting muscle regeneration treats stress urinary incontinence

Wenzhuo Fang[1,3], Xuan Du[2,3], Ranxing Yang[1,3], Meng Liu[1], Ming Yang[1], Yangwang Jin [1], Guo Gao [2] ✉, Qiang Fu[1] ✉ & Ying Wang[1] ✉

Stress urinary incontinence (SUI) remains a significant clinical challenge due to the lack of strategies that simultaneously address muscle degeneration, neurogenic atrophy, and vascular deficits. Here, we report an innovative injectable system that combines a thermo-responsive poly(N-iso-propylacrylamide)-COOH/leucine/decellularized extracellular matrix hydrogel with adipose-derived stem cells (ADSCs) pre-programmed by zeolitic imidazolate framework-8/polyethylene glycol 200@magnesium (ZIF-8/PEG200@Mg) nanoparticles. In vitro, programmed ADSCs exhibit enhanced neurogenic differentiation, while the hydrogels support robust myogenic activity and cell viability. In a female rat model of SUI—chosen to reflect the higher prevalence of SUI in women—the composite system leads to a marked improvement in leak point pressure (LPP) and restores urethral sphincter function. Mechanistic analyses reveals upregulation of muscle regeneration (e.g., *Myoz1*, *Smyd1*) and neurogenesis/neuromuscular junction (NMJ) stabilization genes (e.g., *Dok7*, *Musk*), highlighting a coordinated multi-lineage regenerative process. This work establishes an integrated regeneration-plus-support injectable strategy, offering a regenerative medicine-based approach that surpasses conventional bulking or sling therapies for SUI.

As a subtype of urinary incontinence, stress urinary incontinence (SUI) is characterized by the involuntary leakage of urine in response to increased intra-abdominal pressure, such as during physical activity or coughing. Currently, SUI has an estimated prevalence ranging from 10% to 50% and predominantly affects women over the age of 40 years[1,2]. It markedly impairs quality of life and imposes a substantial healthcare burden worldwide. The pathogenesis SUI is primarily

associated with age-related declines in urethral sphincter function, a substantial decrease in estrogen levels after menopause, and trauma-induced damage to muscle and nerve fibers (such as during childbirth). These factors collectively contribute to the localized atrophy of sphincter musculature and its neural innervation, ultimately leading to urinary dysfunction[3]. For patients with mild to moderate SUI, con-servative treatments such as supervised pelvic floor muscle training,

[1]Department of Urology, Shanghai Sixth People's Hospital Affiliated to Shanghai Jiao Tong University School of Medicine, Shanghai Jiao Tong University, Shanghai 200233, China. [2]School of Automation and Intelligent Sensing, Shanghai Jiao Tong University, Shanghai 200240, China. [3]These authors contributed equally: Wenzhuo Fang, Xuan Du, Ranxing Yang. ✉e-mail: guogao@sjtu.edu.cn; jamesqfu@126.com; sdzbbswangying@alumni.sjtu.edu.cn

electrical stimulation, and pharmacotherapy are commonly recommended in clinical practice[4]. For severe SUI, mid-urethral sling surgery is the primary surgical intervention[5]. Nevertheless, the efficacy and safety of sling procedures are influenced by the properties of the implanted materials and surgical techniques; these may lead to complications such as mesh erosion, local inflammatory responses, infection, and hemorrhage[6]. In recent years, bulking agent injection therapy has garnered increasing attention as the least invasive approach for SUI treatment, particularly for older female patients seeking low-risk surgical interventions[7,8]. However, although these corrective surgical strategies have demonstrated varying degrees of efficacy, they primarily focus on restoring sub-urethral support and fail to effectively address the fundamental pathophysiology of SUI: namely, progressive atrophy of the neuromuscular components of the urethral sphincter[9].

With recent advances in fundamental research, bioengineered approaches that integrate biocompatible hydrogels with muscle cells have been explored to enhance periurethral tissue integrity and promote sphincter regeneration, thus effectively improving urinary function[10,11]. Our previous studies have demonstrated that injecting fragmented extracellular matrix (ECM) derived from decellularized autologous adipose-derived stem cells (ADSCs) sheets into the urethral sphincter of SUI rat models not only provides a bulking effect but also induces muscle regeneration[12]. Owing to their favorable biological characteristics—such as low immunogenicity, stable proliferative capacity, and multilineage differentiation potential—ADSCs have emerged as a promising cell source widely utilized in tissue engineering and regenerative medicine applications[13,14]. By leveraging both cell sheet and decellularized technologies, these ADSCs sheets retain a rich ECM and multiple bioactive cytokines, thereby offering a regenerative strategy for SUI treatment. Notably, in addition to muscular atrophy, the degeneration of periurethral nerves is a critical contributor to urethral sphincter dysfunction and the resulting involuntary urination in SUI[15,16]. It is well established that neural input plays a pivotal role in muscle physiology, not only by inducing muscle contraction but also by exerting trophic effects on muscle cells. These effects include the regulation of muscle metabolism and the secretion of neurotrophic factors that support muscle fiber survival and regeneration[17,18]. Moreover, the neuromuscular junction (NMJ)—a specialized synapse in which motor neurons interface with muscle fibers—is essential for maintaining muscle tone and functional activity[19,20]. In SUI patients, atrophy of urethral sphincter innervation is often accompanied by NMJ impairment, which then disrupts neuromuscular signaling, diminishes muscle tone, and ultimately exacerbates muscle atrophy[21,22]. Therefore, although targeting sphincter regeneration alone can partially restore its function and provide symptomatic relief of SUI, its long-term therapeutic efficacy remains uncertain. This is primarily because of persistent neurodegeneration, which, if left unaddressed, may lead to progressive sphincter dysfunction and recurrent urinary incontinence. Although some recent studies have focused on neuromuscular regeneration, most of these efforts have been directed toward more extensively studied skeletal muscles (such as those in the limbs) or cardiac muscle and its associated innervation[23–25]. By contrast, the unique regenerative challenges of the urethral sphincter, which exhibits a highly intricate anatomical structure and multifaceted functionality[26,27], remain largely unexplored.

To address the outstanding challenges and fill existing gaps in the field, we herein report a regeneration plus mechanical support injectable strategy for the treatment of SUI. Specifically, we engineer a thermoresponsive hydrogel-based delivery system incorporating programmed stem cells, which not only offers localized structural support to the atrophic urethral sphincter but also simultaneously promotes myogenesis, angiogenesis, and neurogenesis—targeting the core pathological hallmark of sphincter degeneration to enable functional reconstruction and recurrence prevention. From a materials design perspective, we synthesize carboxyl-modified PNIPAm (PNIPAm-C) hydrogel by copolymerizing N-isopropylacrylamide (NIPAm) with tert-butyl acrylate (t-BA), followed by trifluoroacetic acid-mediated deprotection. Leucine, a known activator of the mTOR pathway with anabolic effects on muscle protein synthesis and inhibitory effects on proteolysis, was covalently grafted onto PNIPAm-C via EDC/NHS crosslinking to endow the hydrogel with intrinsic myogenic bioactivity[28,29]. To further enrich the local biochemical microenvironment, decellularized ECM (dECM) powders derived from ADSCs sheets were integrated into the hydrogel network. The resulting thermosensitive injectable system employed both hydrophobic associations and amide bond-mediated dual crosslinking to form a mechanically robust and biologically active matrix. Recognizing the limited neuroregenerative capacity of conventional stem cell therapies, metal-organic framework (MOF) nanoparticles were introduced to pre-program stem cells. We chose ZIF-8, which has satisfied biocompatibility and stability[30,31], as the base material and loaded magnesium ions ($Mg^{2+}$) to enhance its neural repair and angiogenesis functions[32,33]. Considering the unstable release problem that might be triggered by the direct incorporation of MOF particles into the hydrogel, we pre-delivered ZIF-8/PEG200@Mg nanoparticles into the interior of ADSCs cells to achieve stem cell pro-vascularization and pro-neurogenesis functional programming before co-injecting them with the hydrogel, which ensured the sustained and stable biological effects.

Overall, the treatment of SUI remains a complex and multifactorial challenge. In a rat model of SUI, localized injection of the engineered injectable composite system into the periurethral region at the bladder neck significantly restored urethral sphincter function, as evidenced by improved LPP at both 4- and 8-weeks post-treatment. Histological analyses further demonstrated robust regeneration of both striated and smooth muscle fibers, accompanied by enhanced vascularization, nerve fiber restoration, and reformation of NMJs. RNA-sequencing (RNA-seq), qRT-PCR, and western blot (WB) analysis revealed that leucine and dECM powder effectively upregulated myogenic markers such as *Myoz1* and *Smyd1*, while preprogrammed ADSCs significantly enhanced expression of neuroregenerative and NMJ-stabilizing genes, including *Ache*, *Dok7*, and muscle-specific kinase (*Musk*). Collectively, this study introduces a regeneration plus mechanical support strategy that outperforms conventional approaches based solely on volumetric bulking or static structural reinforcement (Fig. 1). By simultaneously promoting muscle reconstruction, angiogenesis, and neural repair of the urethral sphincter, our work lays the groundwork for a promising regenerative medicine-based therapy with translational potential for SUI patients.

## Results
### Fabrication and characterization of multi-network crosslinked hydrogels
In the present study, we functionalized thermosensitive PNIPAm hydrogel with carboxyl groups, yielding PNIPAm-C. Briefly, NIPAm was polymerized with t-BA to generate a t-BA-modified PNIPAm polymer. The subsequent reaction with trifluoroacetic acid at 25°C for 48 hours resulted in the complete hydrolysis of t-BA, yielding PNIPAm-C and tert-butanol (Fig. 2A). Compared with PNIPAm, the ${}^{1}$H nuclear magnetic resonance spectrum of PNIPAm-C exhibited a novel peak at 10.1 ppm, indicating the successful introduction of carboxyl groups (Fig. 2B). Fourier transform infrared analysis further confirmed this modification, as evidenced by an absorption peak near 1720 cm-1, which corresponds to the stretching vibration of the C = O bond in carboxyl groups (Fig. 2C). Because PNIPAm hydrogels inherently possess both hydrophilic amide groups (−CONH−) and hydrophobic isopropyl groups (C−CH(CH3)2), increasing temperatures lead to a predominance of hydrophobic interactions, which induces a characteristic sol−gel transformation. To determine whether carboxyl

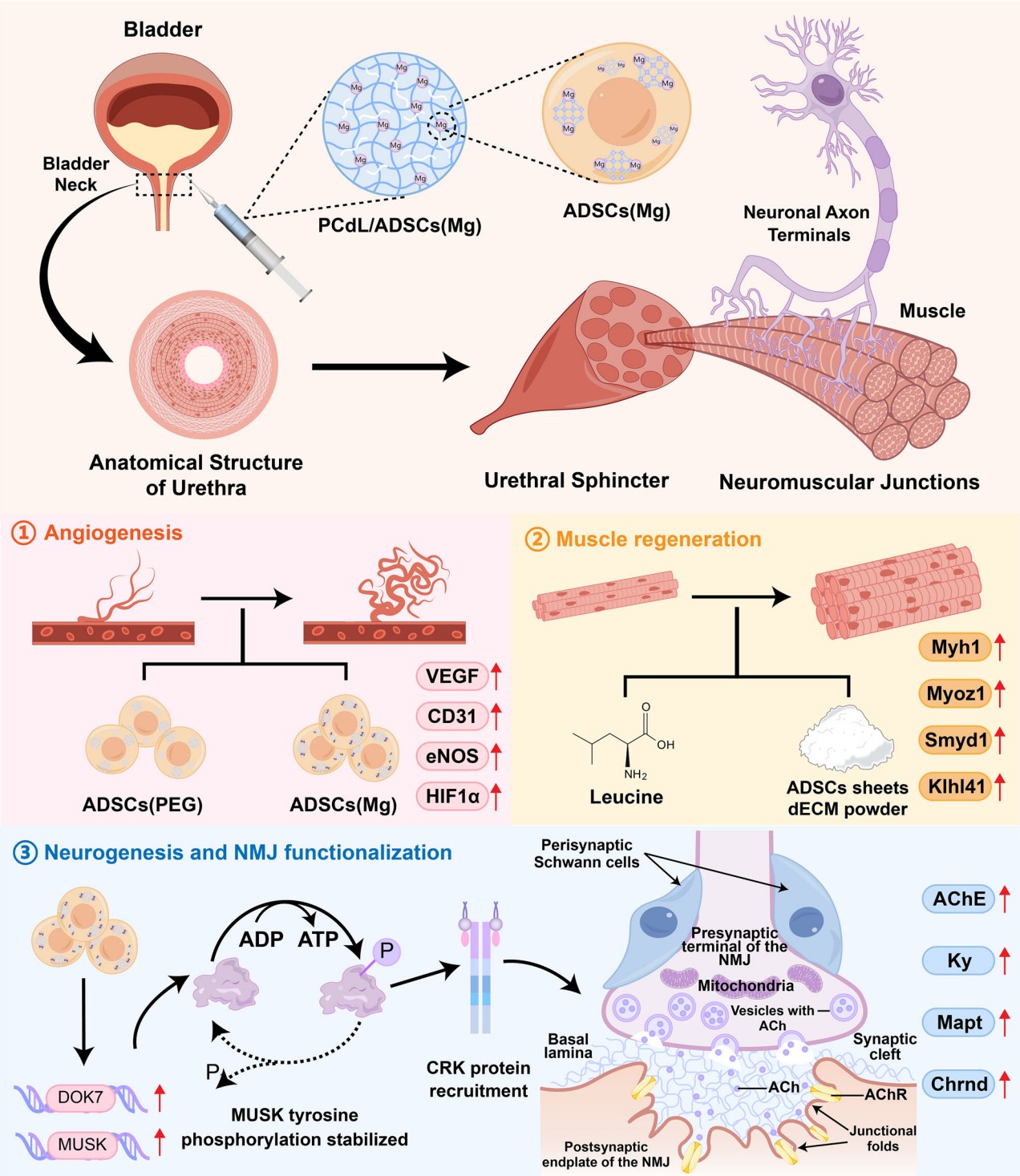

**Fig. 1 | Schematic diagram of injectable therapeutic system loaded with neurogenesis-programmed stem cells for the treatment of SUI.**

functionalization altered the hydrophilic–hydrophobic interactions and thermoresponsive behavior of our hydrogel, we performed rheological analyses. With increasing temperature, the storage modulus of PNIPAm-C progressively increased and surpassed the loss modulus (Fig. 2D, E), thus confirming the sol–gel transition—a critical property for its potential application as an injectable carrier for urethral sphincter treatment. To endow the hydrogel with biological activity and enhance muscle regeneration, we used EDC/NHS coupling to activate the carboxyl groups on PNIPAm-C, subsequently cross-linking it with leucine—widely recognized for its role in promoting

muscle cell regeneration—and the bioactive dECM powder to yield the PCdL hydrogel. As expected, this hydrogel also exhibited sol–gel transformation behavior upon temperature elevation (Fig. 2G, H). Furthermore, differential scanning calorimetry analysis of the two injectable systems revealed exothermic peaks in both PNIPAm-C and PCdL hydrogels (Fig. 2F). This phenomenon likely arose from enhanced hydrophobic interactions between polymer chains at higher temperatures, which then expelled water molecules and induced a more ordered hydrophobic aggregation, thereby releasing energy. To further investigate microstructural changes in the hydrogels, scanning

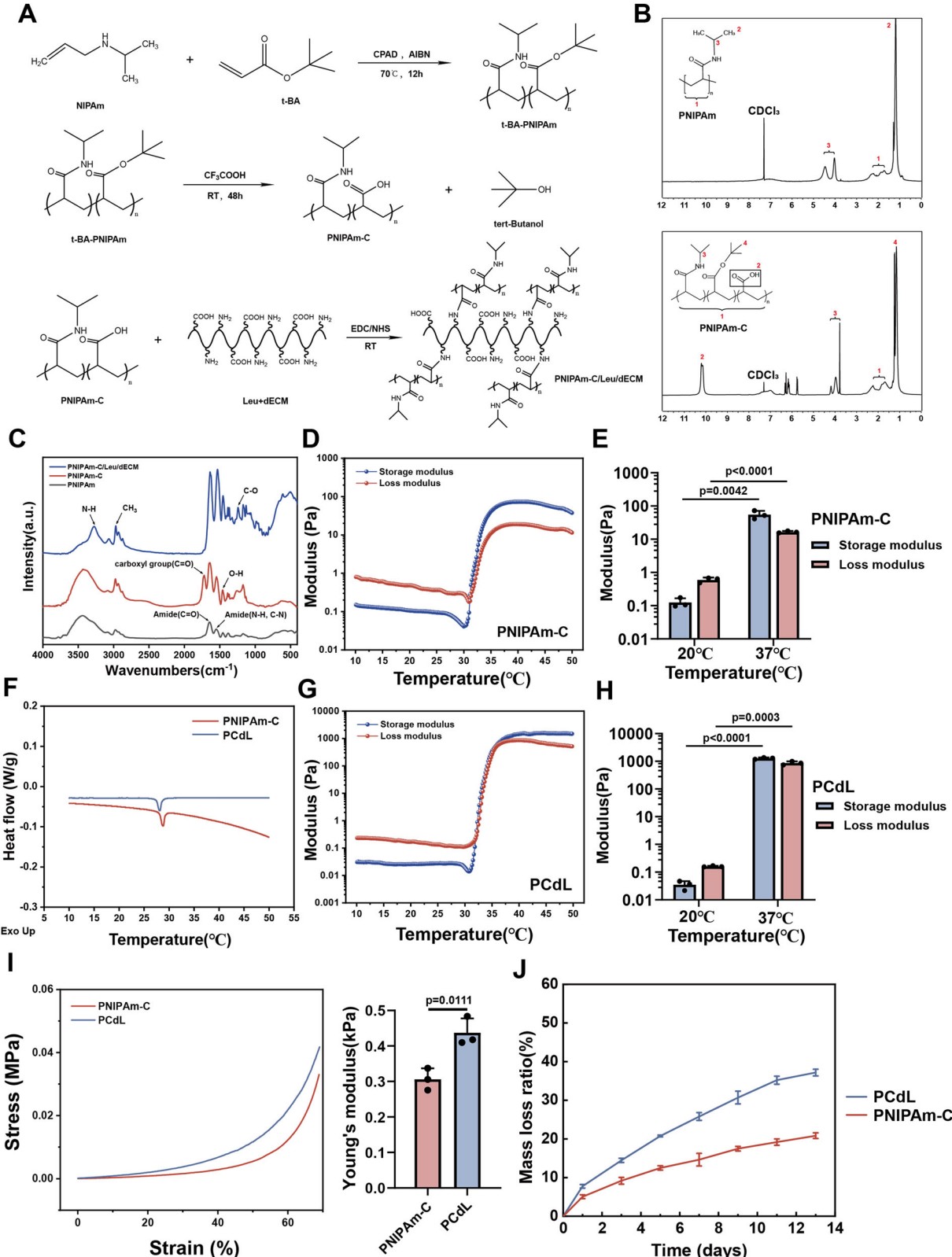

**Fig. 2 | Generation and characterization of hydrogel. A** Schematic of PNIPAm-C/Leu/dECM polymer synthesis: PNIPAm-C was synthesized from NIPAM and t-BA, followed by EDC/NHS-mediated crosslinking with Leu and dECM. **B** 1H NMR spectra of the PNIPAm and PNIPAm-C polymer in CDCl3. **C** FTIR spectra of the different hydrogels (a.u.= arbitrary unit). Temperature-dependent rheology curve of PNIPAm-C (**D**) and PNIPAm-C/Leu/dECM (**G**) at 10–50 °C. **F** DSC thermograms of hydrogels. Comparison of quantitative modulus analysis of PNIPAm-C (**E**) and PNIPAm-C/Leu/dECM (**H**) hydrogels at different temperatures ($n$ = 3 independent samples). **I** Compressive stress-strain curves and Young's modulus of hydrogels ($n$ = 3 independent samples). **J** Degradation rates of hydrogels in PBS ($n$ = 3 independent samples). Data are expressed as mean ± standard deviation (SD). All error bars represent SD. $p$ values calculated using one-tailed unpaired t-test. Source data are provided as a Source Data file.

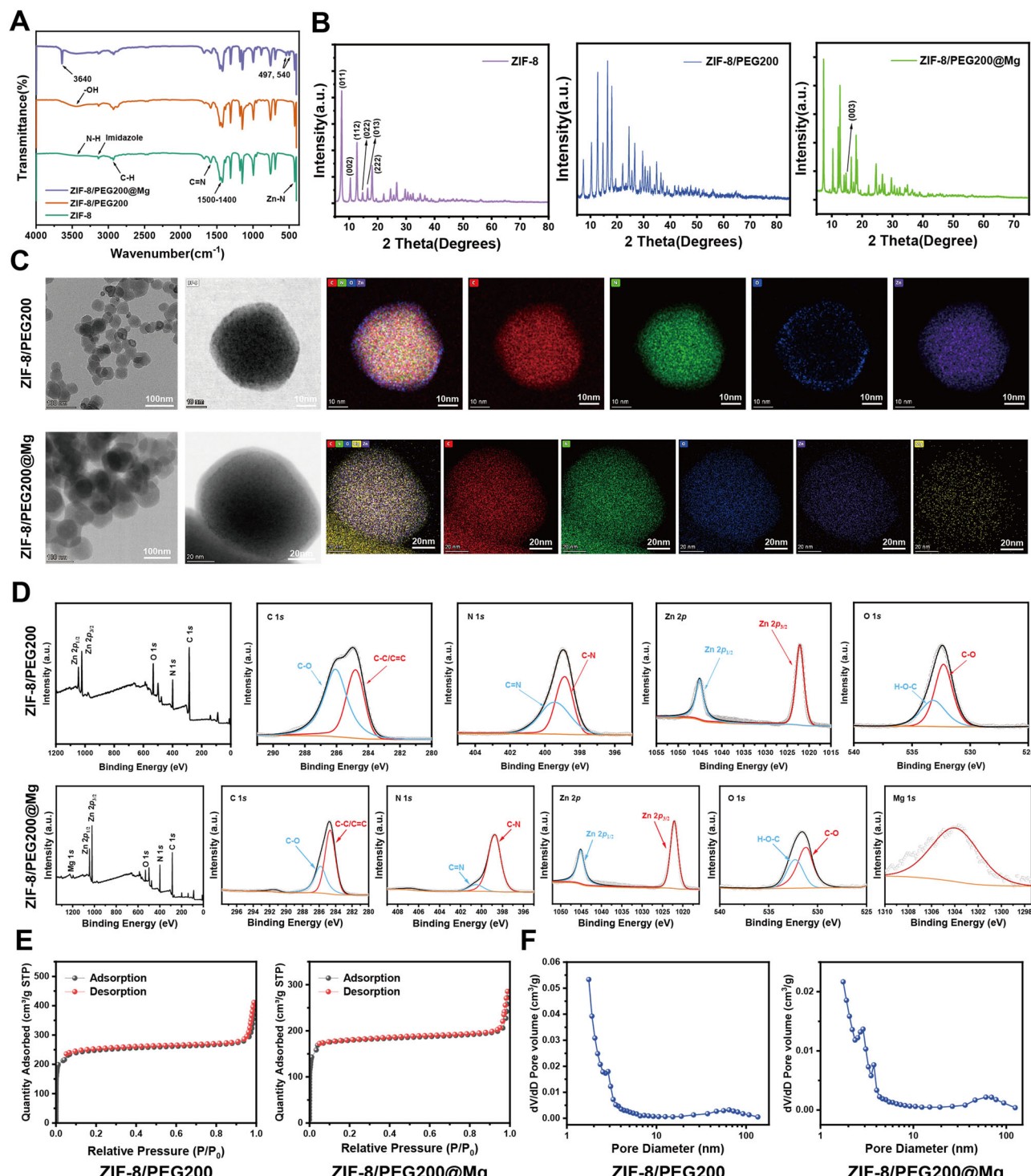

**Fig. 3 | Characterization of various MOF nanoparticles. A** FTIR spectra of ZIF-8, ZIF-8/PEG200 and ZIF-8/PEG200@Mg NPs. **B** XRD characterization of three different MOF NPs (a.u.= arbitrary unit). **C** TEM and EDS elemental mapping of C (red), N (green), O (blue), Zn (purple) and Mg (yellow) for ZIF-8/PEG200 and ZIF-8/PEG200@Mg NPs. Representative image from $n = 3$ biologically independent experiments with similar results. **D** XPS spectra of C 1 s, N 1 s, Zn 2p, O 1 s and Mg 1 s (a.u.= arbitrary unit). **E** N2 adsorption/desorption isotherms of ZIF-8/PEG200 and ZIF-8/PEG200@Mg NPs. **F** Pore width distribution of ZIF-8/PEG200 and ZIF-8/PEG200@Mg NPs. Source data are provided as a Source Data file. Representative image from $n = 3$ independent experiments with similar results.

electron microscopy was used. Both hydrogels exhibited interconnected porous structures (Supplementary Fig. 1); however, the PCdL hydrogel displayed fewer pores with thicker pore walls. This might be attributed to increased crosslinking density caused by the incorporation of leucine and dECM powder. To evaluate the mechanical properties of the gel state, we then compared the Young's modulus

and compressive mechanical strength between the two hydrogel groups. The inclusion of leucine and dECM powder significantly enhanced the mechanical performance and structural stability of the hydrogel (Fig. 2I). Similarly, degradation rate assays revealed that both groups of hydrogels sustained a bulking effect over time. Unexpectedly, the degradation rate of the PCdL hydrogel was higher than that of

the PNIPAm-C group (Fig. 2J); this may be because among the same mass of hydrogels, PCdL has a composition that is complex and rich in hydroxyl, carboxyl, and other degradable groups and substances.

## Characterization of three different MOF NPs

Initially, ZIF-8 NPs were synthesized via the coordination of zinc ions and 2-methylimidazole (2-MIM). However, dynamic light scattering analysis revealed the severe aggregation of ZIF-8 in the absence of ultrasonic dispersion, with an average particle size in the micrometer range and a zeta potential absolute value of $7.89 \pm 1.3$ mV (Supplementary Fig. 2D). To mitigate aggregation and enhance cellular uptake efficiency, surface modification with polyethylene glycol (PEG)200 was performed, yielding ZIF-8/PEG200 MOF particles. As expected, PEGylation significantly reduced the particle size to $242.70 \pm 11.29$ nm while increasing the absolute value of the zeta potential to $27.10 \pm 2.23$ mV (Supplementary Fig. 2A–C), suggesting that the PEG200 coating effectively improved colloidal stability. To further enhance the therapeutic potential for neurogenesis and angiogenesis, ZIF-8/PEG200 was subsequently incubated with magnesium chloride ($MgCl_2$) in an aqueous solution, successfully generating ZIF-8/PEG200@Mg NPs. Fourier transform infrared spectroscopy confirmed these modifications, with all three MOFs exhibiting similar major absorption peaks (Fig. 3A). Compared with ZIF-8, the broad peak at 3550–3345 cm$^{-1}$ corresponded to the stretching vibration of -OH, which was attributed to PEG200 incorporation. Additionally, ZIF-8/PEG200@Mg displayed characteristic Mg-Cl absorption peaks at 497 cm$^{-1}$ and 540 cm$^{-1}$, and a distinct peak at 3640 cm$^{-1}$ indicated the coordination bond formation between $Mg^{2+}$ and $OH^-$. X-ray diffraction analysis further demonstrated that ZIF-8/PEG200 retained the characteristic diffraction peaks of ZIF-8, albeit with slight shifts (Fig. 3B). Notably, ZIF-8/PEG200@Mg exhibited an additional diffraction peak corresponding to the (003) plane of $MgCl_2$, thus confirming the successful incorporation of $Mg^{2+}$.

To elucidate the elemental compositions and spatial distributions of the modified MOFs, transmission electron microscopy coupled with elemental mapping was performed. Oxygen signals were detected in ZIF-8/PEG200 because of PEGylation, whereas the distributions of carbon, nitrogen, and zinc remained consistent between ZIF-8 and ZIF-8/PEG200 (Fig. 3C, Supplementary Fig. 3C). Importantly, ZIF-8/PEG200@Mg exhibited a uniform distribution of magnesium both on the surface and within the MOF matrix. Consistent with these findings, X-ray photoelectron spectroscopy (*XPS*) spectra further validated the presence of magnesium (Fig. 3D, Supplementary Fig. 3A). As anticipated, the magnesium spectrum displayed prominent peaks exclusively in ZIF-8/PEG200@Mg, whereas oxygen peaks were evident in both ZIF-8/PEG200 and ZIF-8/PEG200@Mg. Scanning electron microscopy analysis further revealed distinct morphological changes among the three MOFs (Supplementary Fig. 3B). ZIF-8 exhibited a characteristic rhombic dodecahedral structure with marked aggregation. Upon PEG200 modification, this aggregation tendency was diminished, leading to a more monodisperse NPs distribution. Following $MgCl_2$ incorporation, ZIF-8/PEG200@Mg transitioned into a more spherical morphology with blurred edges, likely caused by surface interactions with $Mg^{2+}$. Furthermore, nitrogen adsorption–desorption isotherm analysis provided insights into the porous structures of the synthesized MOFs. ZIF-8 displayed a hybrid isotherm combining Type-I and Type-IV characteristics, along with an H1-type hysteresis loop that is typically observed in spherical particle aggregates with uniform mesopores (Supplementary Fig. 4). Brunauer–Emmett–Teller analysis determined the specific surface area of ZIF-8 to be approximately 1,112 m²/g. By contrast, the adsorption curve of ZIF-8/PEG200 exhibited a sharp increase in nitrogen uptake at low relative pressure, reaching saturation at approximately 200 cm³/g, which is characteristic of Type-I isotherms (Fig. 3E, F). Moreover, Brunauer–Emmett–Teller analysis revealed a reduced specific surface area of approximately 778 m²/g. Similarly, ZIF-8/PEG200@Mg

exhibited an adsorption–desorption profile that resembled that of ZIF-8/PEG200, with a further reduction in specific surface area to approximately 549 m²/g, indicating that structural modifications may be induced by $Mg^{2+}$ incorporation.

## ZIF-8/PEG200@Mg NPs achieve neurogenic programming of ADSCs

In our previous studies, we have demonstrated that injecting fragmented ADSCs sheets into the urethral sphincter promotes muscle regeneration. The atrophy of sphincter-associated nerves is a crucial pathological hallmark of SUI. We therefore attempted to integrate muscle regeneration and neurogenesis within the urethral sphincter to enhance the therapeutic efficacy of the injection system against SUI. In the present study, we used MOF NPs to program ADSCs. We then performed RNA sequencing on cell clusters 3 days post-MOF NPs treatment to identify the differentially expressed genes (DEGs), signaling pathways, and underlying biological processes of cellular reprogramming. To determine the optimal MOF NPs concentration for ADSCs programming, we conducted Cell Counting Kit 8 (CCK8) assays to assess cell viability following exposure to various concentrations of MOF NPs. Even at a concentration of 40 µg mL-1, ZIF-8/PEG200 and ZIF-8/PEG200@Mg NPs did not compromise ADSCs viability (Supplementary Fig. 5). Furthermore, bio-transmission electron microscopy demonstrated the successful internalization of MOF NPs within ADSCs (Supplementary Fig. 6). Heatmaps of sample similarity, principal component analysis, and heatmaps of DEG expression profiles revealed distinct transcriptomic differences between the groups (Fig. 4A; Supplementary Fig. 7A, B). Additionally, volcano plots and Venn diagrams further underscored the significant effects of MOF NPs on ADSCs gene expression (Supplementary Fig. 7C, Supplementary Fig. 8).

To gain deeper mechanistic insights, we performed Kyoto Encyclopedia of Genes and Genomes (KEGG) and Gene Ontology (GO) enrichment analyses of the most upregulated genes (Fig. 4B–D). Compared with untreated ADSCs, ZIF-8/PEG200-treated ADSCs exhibited significant enrichment in ECM secretion, cell migration, and zinc ion response pathways. KEGG pathway enrichment also suggested that ZIF-8/PEG200 internalization might trigger immune responses, potentially because of the recognition of MOF NPs as foreign matter. Surprisingly, ADSCs treated with ZIF-8/PEG200@Mg displayed significantly upregulated pathways associated with neural regeneration, neurogenesis, and actin cytoskeleton organization. Specifically, pathways such as mineral absorption, regulation of actin cytoskeleton, and axon guidance were highly expressed, suggesting that ZIF-8/PEG200@Mg effectively reprograms ADSCs toward a neurogenic phenotype. To further elucidate the molecular mechanisms underlying ZIF-8/PEG200@Mg-mediated ADSCs reprogramming, we performed qRT-PCR and WB analyses to evaluate key gene expression changes related to the ECM (*Itga2* and *Clasp2*), neural system (*Atxn7*, *Kirrel3*, *Nptxr*, *Fez1*, *Ntn1*, and *Plxnb1*), and actin cytoskeleton (*Wasf2* and *Acta2*) (Fig. 4E, F). Notably, emerging studies have identified SHANK3 protein as a critical regulator of NMJ formation. SHANK3 deficiency induces synaptic dysfunction, thus impairing neuronal connectivity and leading to peripheral neuromuscular structural defects that ultimately result in muscle dysfunction[34]. Strikingly, our results revealed a significant upregulation of SHANK3 in ADSCs following ZIF-8/PEG200@Mg internalization. Given these findings, we hypothesize that injecting programmed ADSCs into the periurethral sphincter microenvironment may facilitate local muscle function restoration, NMJ repair, and neurogenesis, thereby offering a promising strategy for SUI treatment.

## ADSCs programmed by MOF NPs promote angiogenesis

Zinc and magnesium ions are essential metal ions in the human body; they promote endothelial proliferation and migration and

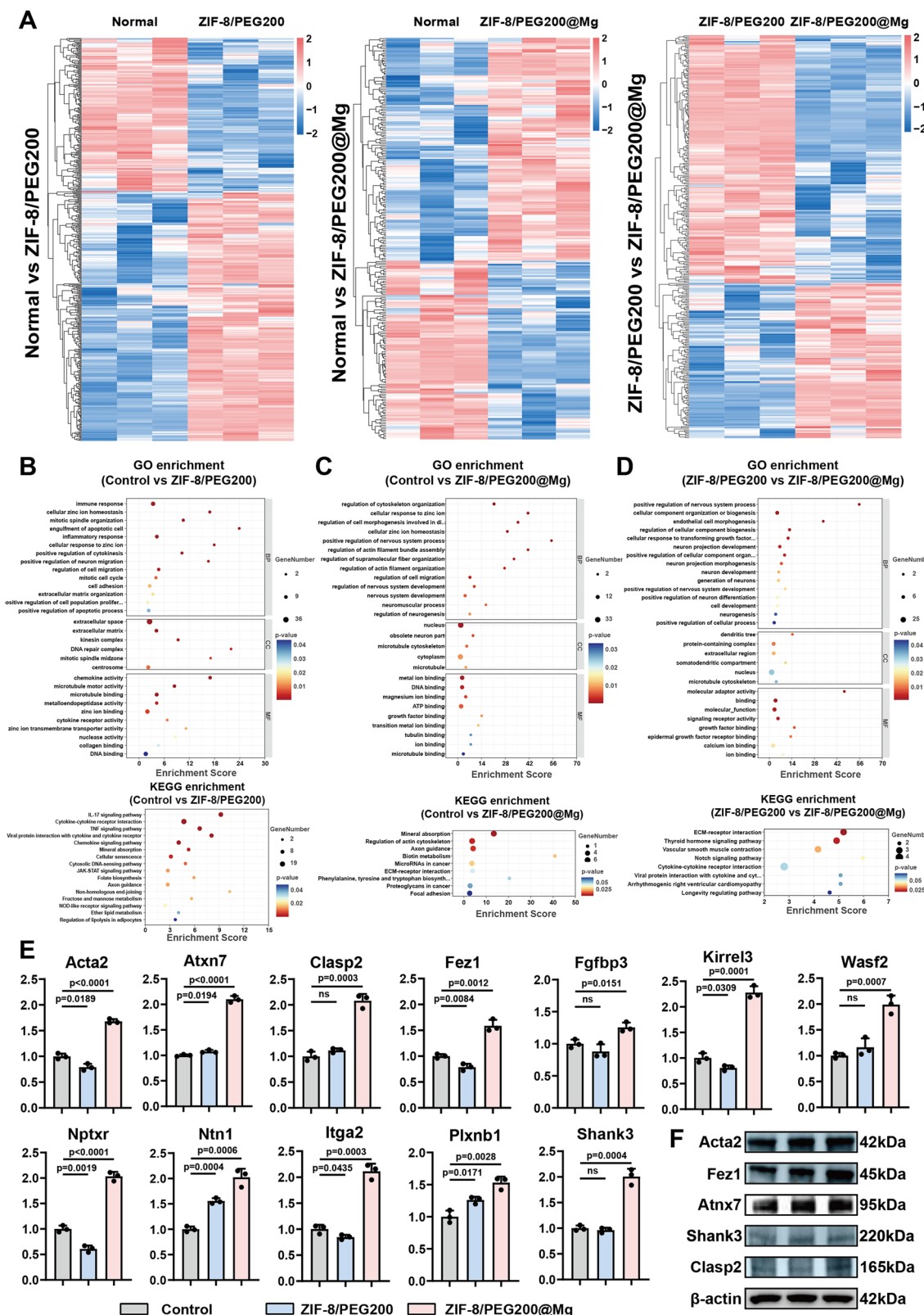

angiogenesis-related gene expression[35,36]. However, most studies have primarily focused on increasing extracellular ion concentrations to achieve these effects. To explore whether MOF NPs-programmed ADSCs enhance angiogenesis and endothelial cell function, we co-cultured different ADSCs with human umbilical vein endothelial cells (HUVECs) (Fig. 5A). Tube formation assays revealed that co-culturing untreated ADSCs with HUVECs moderately improved vascular network formation, likely because ADSCs-secreted growth factors and nutrients enhanced HUVEC viability. Furthermore, HUVECs co-cultured with MOF NPs-programmed ADSCs exhibited significantly enhanced tube formation, and the ZIF-8/PEG200@Mg-programmed ADSCs (ADSCs [Mg]) group displayed the most pronounced angiogenic capacity (Fig. 5B, C). To further assess the effects of different ADSCs on endothelial migration, we performed a wound healing assay. MOF NPs-programmed

**Fig. 4 | Exploration of the effects of MOF NPs on ADSCs programmed by sequencing. A** Heat map analysis of differential genes for two-by-two comparisons of ADSCs clusters in the untreated group (normal), ZIF-8/PEG200 NPs treated group (ZIF-8/PEG200) and the ZIF-8/PEG200@Mg NPs treated group (ZIF-8/PEG200@Mg). **B** Up-regulated pathways in GO and KEGG enrichment analysis after mRNA sequencing between normal and ZIF-8/PEG200 group. **C** Up-regulated pathways in GO and KEGG enrichment analysis after mRNA sequencing between normal and ZIF-8/PEG200@Mg group. **D** Up-regulated pathways in GO and KEGG enrichment analysis after mRNA sequencing between ZIF-8/PEG200 and ZIF-8/PEG200@Mg group. **E** Expression of genes related to extracellular matrix, nervous system and actin of normal, ZIF-8/PEG200 and ZIF-8/PEG200@Mg group via qRT-PCR tests ($n = 3$ independent samples). **F** Representative WB images of ADSCs after treatment of MOF NPs for 3 days. Data are expressed as the mean ± SD. All error bars represent SD. $p$ values calculated using one-tailed unpaired t-test. ns, not significant ($p > 0.05$). Source data are provided as a Source Data file.

ADSCs significantly promoted HUVECs migration, with ADSCs (Mg) exhibiting the strongest effect (Fig. 5D). Furthermore, we conducted qRT-PCR to analyze the expression of angiogenesis-related genes in HUVECs following 7 days of co-culture with different ADSCs (Fig. 5E). Consistent with the tube formation and wound healing assay results, HUVECs exhibited upregulated expression of key angiogenic genes. Collectively, these findings indicate that the injection of MOF NPs-programmed ADSCs into the periurethral sphincter microenvironment holds promise for enhancing local angiogenesis, thereby facilitating tissue repair and restoring sphincter function.

### Characterization of ADSCs sheets with dECM and evaluation of the biocompatibility of the injection system
Intact ADSCs sheets were able to be retrieved from culture dishes using forceps (Supplementary Fig. 9 and Supplementary Movie 1). Subsequently, we subjected the ADSCs sheets to a decellularization process and observed a substantial reduction in DNA content; however, the levels of essential growth factors, including vascular endothelial growth factor (VEGF), platelet-derived growth factor-BB (PDGF-BB), epidermal growth factor (EGF), and hepatocyte growth factor (HGF), remained largely unaffected (Supplementary Figs. 10 and 11). This finding indicates that the decellularization process efficiently removes nuclear components while preserving bioactive factors that are crucial for stem cell function. The decellularized ADSCs sheets were then lyophilized and ground at low temperatures to obtain dECM powder.

To evaluate the cytocompatibility and biological functionality of the injectable system, we co-cultured ADSCs with three formulations: PNIPAm-C polymer (PC), PNIPAm-C + leucine polymer (PC/Leu), and PNIPAm-C + leucine + ADSCs sheets-derived dECM powder polymer (PC/Leu/dECM). Live/dead staining confirmed the satisfied biocompatibility of all three formulations (Fig. 5F; Supplementary Fig. 12). Furthermore, the ADSCs successfully adhered and remained viable in both the PC and PC/Leu/dECM hydrogels, with a uniform distribution and high cell viability (Fig. 5G). These findings highlight the feasibility of using the injectable system to deliver programmed ADSCs for periurethral sphincter therapy. CCK8 assays further confirmed the favorable biocompatibility of the PC/Leu/dECM hydrogel, with dECM significantly enhancing cell proliferation (Fig. 5H). Similarly, scratch wound assays revealed that dECM enhanced cell migration; ADSCs cultured within the PC/Leu/dECM hydrogel achieved nearly complete wound closure within 48 hours (Fig. 5I, J; Supplementary Fig. 13), whereas the coverage rates of other groups remained below 60%.

### Evaluation of an injectable system to restore urethral sphincter function in SUI rats
To further evaluate the therapeutic efficacy of the injectable system loaded with neurogenesis-programmed stem cells for restoring urethral sphincter function and treating SUI, we established a rat SUI model. Briefly, 4-week-old female rats underwent vaginal balloon dilation combined with bilateral ovariectomy to mimic the two primary etiological factors of female SUI: trauma-induced injury to the pelvic floor muscles and nerves, and estrogen deficiency post-menopause. Four weeks post-surgery, different injectable formulations—including saline (SUI group), PNIPAm-C hydrogel (PC group),

PNIPAm-C + leucine + ADSCs sheet-derived dECM powder (PCdL group), PCdL + ADSCs + ZIF-8/PEG200 (PCdL/ADSCs [PEG] group), and PCdL + ADSCs + ZIF-8/PEG200@Mg (PCdL/ADSCs [Mg] group)—were administered around the urinary sphincter for therapeutic intervention. Given the inherent timeline required for muscle regeneration, functional recovery, and neurogenesis, LPP measurements and histological evaluations of the urethral sphincter were performed at weeks 4 and 8 post-injection (Supplementary Fig. 14). A schematic timeline illustrating the application of the injectable system in the rat SUI model and its corresponding efficacy assessment is shown in Fig. 6A. At week 4 post-injection, LPP in the PCdL/ADSCs (Mg) group was restored to near-normal levels ($P > 0.05$) and was significantly higher than that in the SUI group (Fig. 6B). Notably, the PC group also exhibited a slightly higher LPP than that in the SUI group, which was potentially caused by the bulking effect of the PC hydrogel. At week 8, LPP in the PCdL/ADSCs (Mg) group remained elevated and was slightly higher than that at week 4 (Fig. 6C). This sustained improvement may be attributed to the prolonged neurogenic effects and NMJ restoration facilitated by ZIF-8/PEG200@Mg-programmed ADSCs and further validates the long-term therapeutic efficacy of the PCdL/ADSCs (Mg) injectable system.

### Evaluation of muscle regeneration and neurogenesis in the urethral sphincter following injectable treatment
Next, we performed histological staining to evaluate the local tissue conditions and muscle regeneration levels in the urethral sphincter at the bladder neck. Hematoxylin and eosin staining revealed severe atrophy of the urethral sphincter in SUI rats, which was characterized by loose, disorganized tissue and thin, irregular muscle fibers (Fig. 6D). Moreover, the muscle staining appeared lighter and the muscle fibers exhibited faded coloration. In the PC group, no marked sphincter regeneration was observed; there was a reduction in muscle fiber area, exhibiting small, degenerative structures with irregular fiber alignment. By contrast, in the PCdL group (which contained leucine and abundant cytokines), muscle atrophy was alleviated and there were larger and more organized muscle fibers with increased density, aligning with the LPP measurement results. Furthermore, the PCdL/ADSCs (PEG) and PCdL/ADSCs (Mg) groups exhibited substantial muscle regeneration and were nearly indistinguishable from the normal group. Myofibrils appeared red or pink and the nuclei were located at the periphery of muscle fibers, displaying an elongated or elliptical morphology. The muscle fibers exhibited well-defined boundaries, compact structures, and regular morphology. Compared with the staining results at week 4 post-treatment, the urethral sphincter at week 8 demonstrated even more pronounced muscle regeneration. Masson's trichrome staining revealed a similar trend (Fig. 6E). The SUI and PC groups exhibited pale muscle fibers that appeared light red, indicative of small, degenerative muscle fibers. Additionally, following muscle atrophy, collagen fibers accumulated between the muscle fibers, thus enlarging fiber gaps; connective tissue then infiltrated, partially replacing the muscle fibers and altering tissue structure. By contrast, the PCdL group exhibited distinct, deeply stained muscle fibers, and smooth muscle regeneration was more pronounced than that of striated muscle. In the PCdL/ADSCs (Mg) group, the muscle

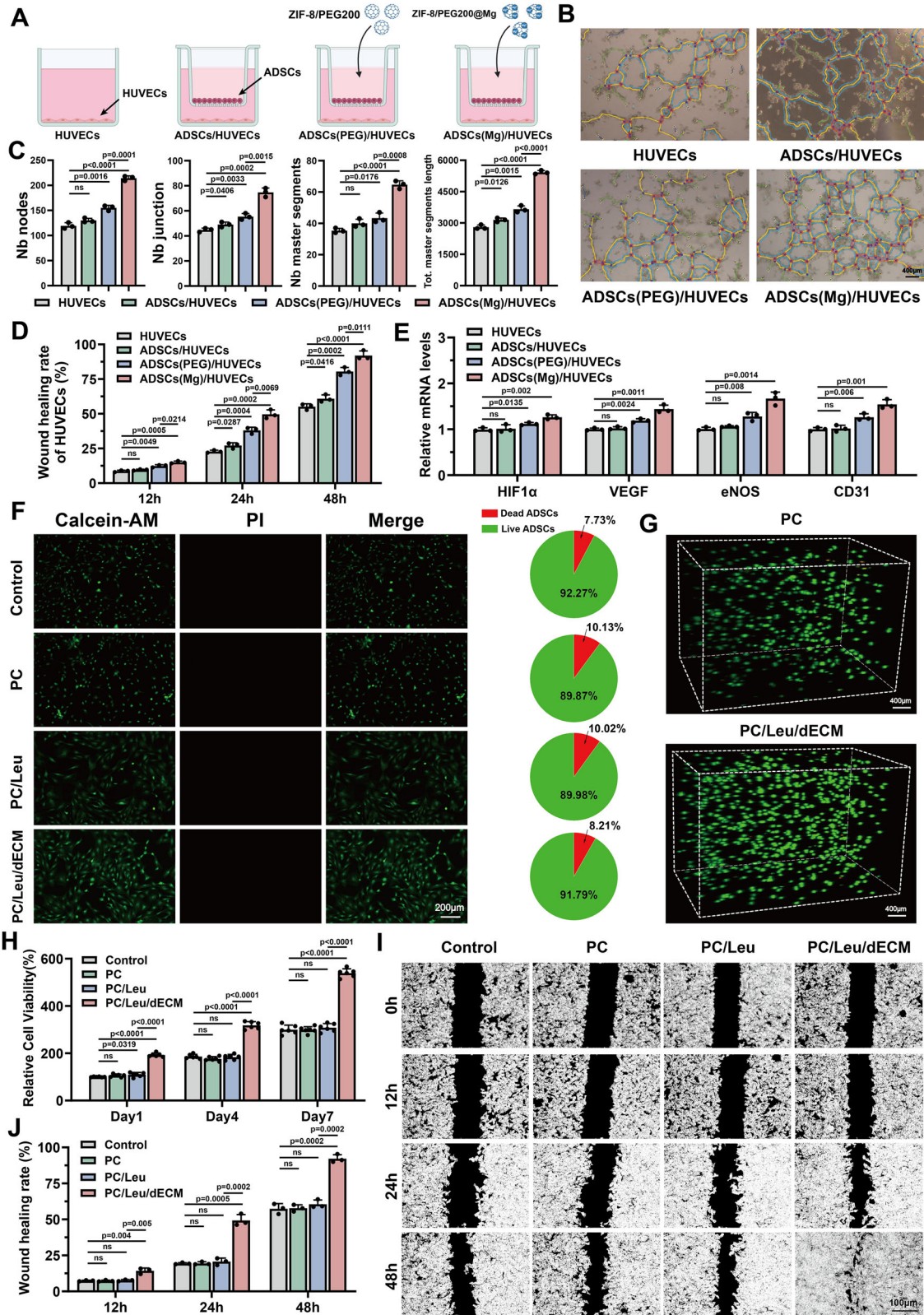

fibers appeared bright red and had increased fiber diameters, compact alignment, and minimal connective tissue infiltration.

With an aim to elucidate the therapeutic effects of the injectable therapeutic system loaded with neurogenesis-programmed ADSCs, we used immunofluorescent staining to track desmin, α-smooth muscle actin (α-SMA), synaptophysin, tubulin β3 (TUBB3), and VEGF, which indicate muscle regeneration, synaptogenesis,

neurogenesis, and angiogenesis, respectively. Both desmin and α-SMA expression levels were markedly lower in the SUI and PC groups, which was consistent with the hematoxylin and eosin and Masson's trichrome staining results and indicates severe muscle atrophy. Although fluorescent intensity was significantly elevated in the PCdL group, it remained lower than that observed in the PCdL/ADSCs (PEG) and PCdL/ADSCs (Mg) groups, suggesting that

**Fig. 5 | Validation of the ADSCs to promote vascularization after MOF NPs programming and detection of biocompatibility and biological function of the injection system. A** Schematic illustration of the co-culture system. ADSCs with different treatments are located in the upper chamber and HUVECs in the lower chamber. Elements were created in BioRender. Wenzhuo, F. (2025) https://BioRender.com/j76b692. **B**, **C** The tube formation ability of HUVECs in different group (scale bar = 400 μm) ($n$ = 3 biologically independent samples). **D** The migration ability of HUVECs culture with or without a co-culture system was measured by wound healing assay ($n$ = 3 biologically independent samples). **E** Genes associated with angiogenesis expression levels of HUVECs in different group after 7-days co-culture ($n$ = 3 biologically independent samples). **F** Fluorescence microscopy pictures and semi-quantitative data demonstrating live (green) and dead (red) cells after calcein-AM (green) and PI (red) staining in different injection system (scale bar = 200 μm). Representative image from $n$ = 3 biologically independent experiments with similar results. **G** 3D reconstruction of live/dead-stained ADSCs in PC and PC/Leu/dECM injection system (scale bar = 400 μm). **H** Cell proliferation rates of ADSCs at 1-, 4- and 7-days post-treatment were detected with the CCK8 assay ($n$ = 6 biologically independent samples). **I** Dark-field images and (**J**) corresponding quantitative data of ADSCs cultured under different conditions after 12 h, 24 h and 48 h ($n$ = 3 biologically independent samples) (scale bar = 100 μm). Data are expressed as mean ± SD. All error bars represent SD. $p$ values calculated using one-tailed unpaired t-test. ns, not significant ($p > 0.05$). Source data are provided as a Source Data file.

MOF-programmed ADSCs promote the regeneration of both smooth and striated muscle within the urethral sphincter. Notably, no significant differences in local muscle regeneration were observed between ZIF-8/PEG200- and ZIF-8/PEG200@Mg-programmed ADSCs (Fig. 7A, B, E, F). However, synaptophysin and TUBB3 staining underscored the superior neurogenic potential of ZIF-8/PEG200@Mg NPs. The fluorescent intensity in the PCdL/ADSCs (Mg) group was markedly higher than that in the PCdL/ADSCs (PEG) group and was close to that in the normal group. This finding suggests that ADSCs programmed by ZIF-8/PEG200@Mg NPs exhibit potent neurogenic and synaptogenic capabilities. These may then facilitate the formation of neuromuscular and synaptic networks that provide essential trophic and functional support to the regenerating urethral sphincter, thereby further restoring sphincter function (Fig. 7C, G; Supplementary Fig. 15; Supplementary Fig. 17B). Moreover, VEGF staining highlighted the capacity of the injectable system to promote angiogenesis within muscle tissue; the highest fluorescent intensity was observed in the PCdL/ADSCs (Mg) group. This finding was consistent with the in vitro co-culture results of different ADSCs groups with HUVECs, indicating that MOF-programmed ADSCs effectively enhance in vivo muscle tissue vascularization (Supplementary Figs. 16 and 17A).

To comprehensively assess neuromuscular innervation, NMJ regeneration, and distribution in the PCdL/ADSCs (Mg) group, we further performed immunohistochemical staining for acetylcholinesterase (Ache) (Fig. 7D, H). The NMJ serves as the contact point between motor neuron terminals and muscle fibers and plays a pivotal role in triggering muscle contraction, maintaining muscle tone, and preventing muscle atrophy. Given that NMJ represent specialized chemical synapses that use acetylcholine as a neurotransmitter, Ache—a key enzyme in neuromuscular signaling—functions to degrade synaptic acetylcholine and serves as a biomarker for NMJ localization. Immunohistochemical analysis revealed an almost complete absence of Ache expression in the SUI and PC groups. By contrast, the PCdL/ADSCs (Mg) group exhibited the highest Ache expression and displayed intense local positivity, particularly in the post-synaptic membrane and basal lamina regions, forming distinct granular or band-like staining patterns. Moreover, analysis of NMJ occupancy serves as a quantitative indicator of neuromuscular junction innervation and offers sensitive resolution for evaluating the extent of nerve regeneration. In this study, triple immunostaining for NMJs and axons revealed a markedly higher occupancy rate in the PCdL/ADSCs (Mg) group at 8 weeks post-injection, indicating enhanced NMJ maturation and neuroregenerative outcomes (Supplementary Figs. 18 and 19).

In summary, the PCdL/ADSCs (Mg) therapeutic system not only facilitated urethral sphincter muscle regeneration and angiogenesis but also demonstrated great potential for promoting neurogenesis, NMJ formation, and functional restoration, thereby playing a pivotal role in the treatment of SUI in rats.

## Exploration of the deeper mechanisms underlying the injectable system that promote muscle regeneration and neurogenesis in the treatment of SUI

To further elucidate the molecular mechanisms by which the injectable system promotes urethral sphincter regeneration and neurogenesis, we conducted RNA-seq on urethral tissues harvested seven days post-injection. DEG analysis was performed to decode biological processes involved in tissue repair. A heatmap of DEGs (Fig. 8A; Supplementary Fig. 21A) visually highlighted widespread transcriptional alterations across treatment groups versus the SUI control, with upregulated genes marked in red and downregulated in blue[37]. Principal component analysis (PCA) revealed distinct transcriptomic separation among PCdL-, PCdL/ADSC(PEG)-, and PCdL/ADSC(Mg)-treated groups (Fig. 8B–D; Supplementary Fig. 20A; Supplementary Fig. 21B), indicating that each formulation significantly modulated the gene expression landscape[38]. Complementary volcano plots (Fig. 8B–D; Supplementary Fig. 21C) mapped $\log_2$FC against $-\log_{10}$P-values, further underscoring the intensity and directionality of transcriptional responses induced by each injectable strategy[39]. Notably, DEG quantification (Supplementary Fig. 20B) demonstrated the PCdL/ADSC(Mg) group harbored the highest number of DEGs, suggesting a more robust and comprehensive reprogramming effect on urethral tissues[40]. To dissect the functional implications of upregulated DEGs, we conducted GO and KEGG pathway enrichment analyses (Fig. 8E–G; Supplementary Fig. 21D). In the PCdL group, upregulated genes were predominantly enriched in immune response, muscle contraction, and wound healing, indicating that the hydrogel scaffold alone could elicit localized immunomodulation and tissue remodeling. GO enrichment under Cellular Component terms highlighted associations with striated muscle thin filament, extracellular space, and sarcomere, supporting structural remodeling of contractile apparatuses (Fig. 8E). Concurrently, KEGG analysis revealed activation of proliferative signaling cascades, such as the MAPK pathway, likely mediating hydrogel-driven tissue repair[41]. In the PCdL/ADSC(PEG) group, DEGs were further enriched in muscle cell contraction and muscle fiber development, reflecting the promyogenic influence of stem cell inclusion[37]. Enrichment in muscle myosin complex and sarcomere-associated regions suggests reinforced muscle functionality. In terms of molecular function, FATZ binding and telethonin binding were significantly upregulated. FATZ proteins stabilize Z-disc integrity and mediate mechanical signaling[42], while telethonin is crucial for sarcomere assembly and mechanical signaling transduction[43]—together implying enhanced sarcomeric maturity and contractile readiness (Fig. 8F). Of particular note, the PCdL/ADSC(Mg) system uniquely induced neuroregenerative signatures, including positive regulation of neuron projection development and neuromuscular junction development (Fig. 8G). These neurogenic processes are vital for re-establishing neural circuits and restoring sphincter innervation, distinguishing this group mechanistically from the others[44]. GO cellular

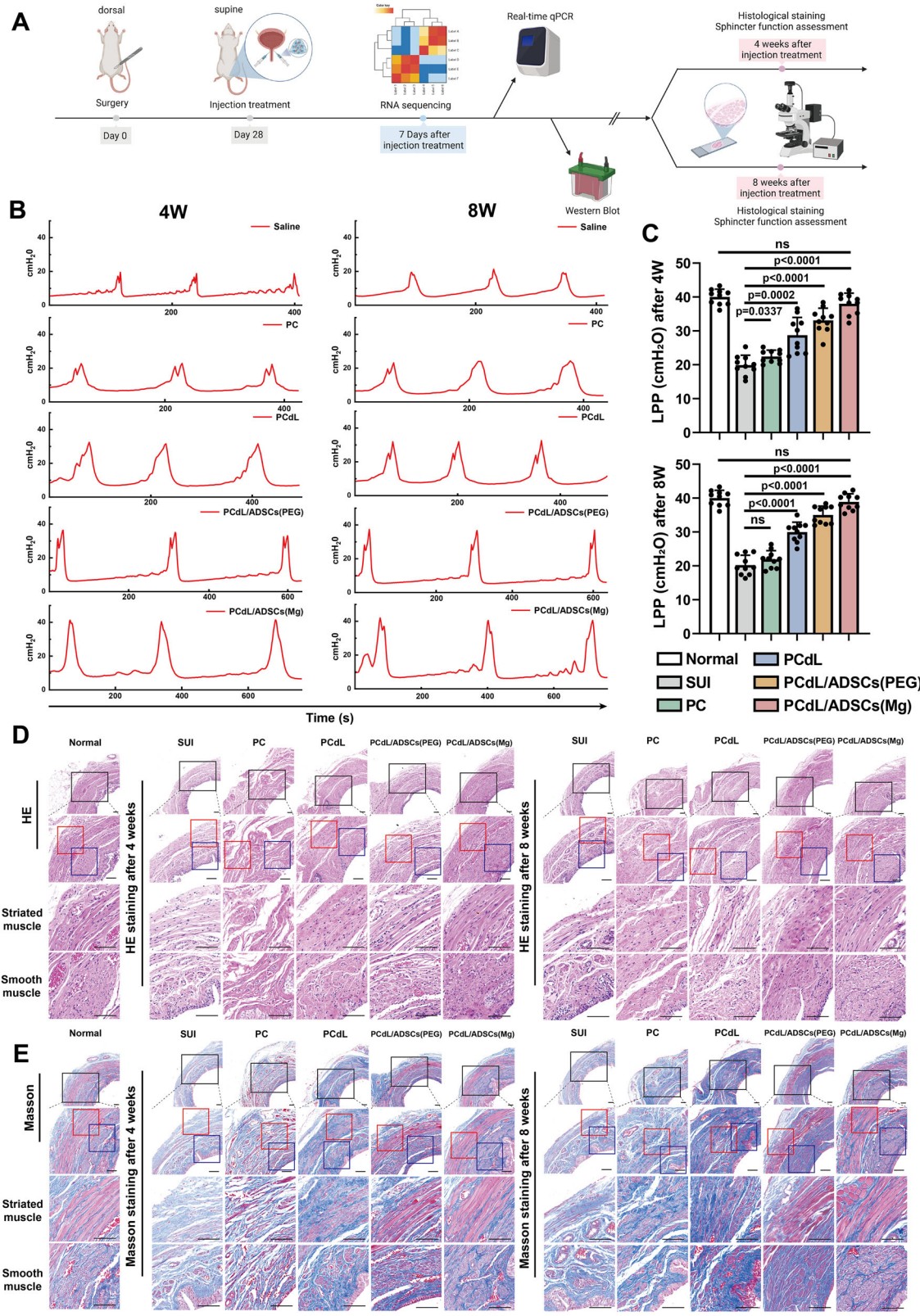

**Fig. 6 | In vivo evaluation of injectable systems for restoring urethral sphincter function and promoting muscle regeneration. A** Timeline for modeling, treating, and detecting efficacy in SUI rats. Elements were created in BioRender. Wenzhuo, F. (2025) https://BioRender.com/n81p676. **B** Curve plotted by LPP values of different groups at different time points. **C** Histograms demonstrating LPP values for different groups at different time points (*n* = 10 biologically independent samples). **D** HE and **E** Masson staining of the urethral sphincter at the bladder neck after 4 and 8 weeks of injection therapy. The black box represents the area of magnified observation, and the red and blue boxes represent striated and smooth muscles, respectively. Scale bars: 100 μm. *n* = 10 biologically independent samples. Data are expressed as the mean ± SD. All error bars represent SD. p values calculated using one-tailed unpaired t-test. ns, not significant (*p* > 0.05). Source data are provided as a Source Data file.

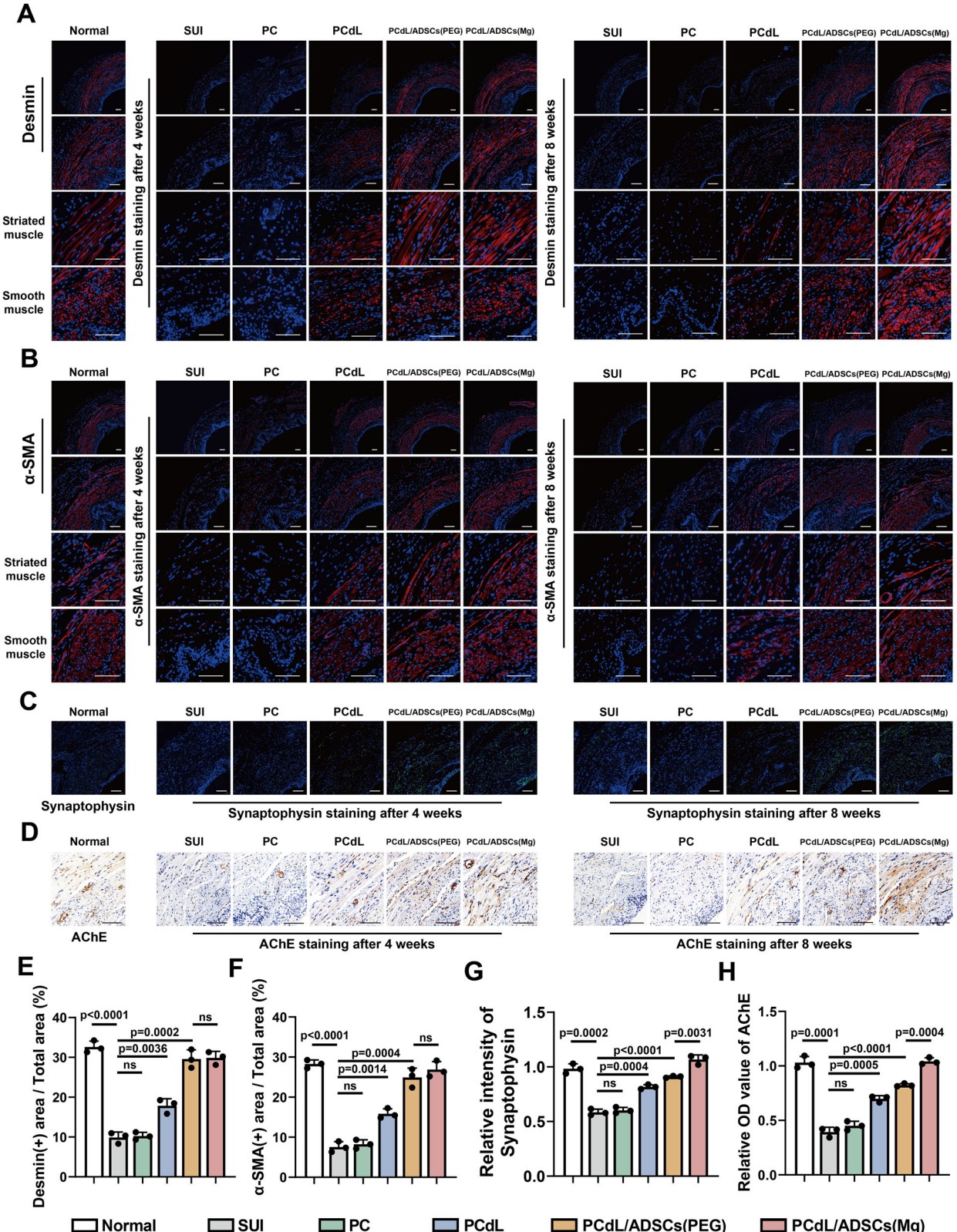

**Fig. 7 | Urethral sphincter tissue sections stained to assess muscle, vascular and neural tissue regeneration after injection treatment.** Immunofluorescence staining images of (**A**) Desmin, **B** α-SMA and (**C**) Synaptophysin. Blue color represents the nucleus, and red (Desmin/α-SMA) and green (Synaptophysin) represent the corresponding markers. **D** Immunohistochemical staining images of Ache. scale bars: 100 μm. Comparison of (**E**) Desmin and (**F**) α-SMA positive areas to the total area, **G** relative fluorescence intensity of Synaptophysin, and (**H**) relative OD value of Ache of urethral sphincter after 4 weeks of treatment with different injection systems. Data are expressed as the mean ± SD ($n = 3$ biologically independent samples). All error bars represent SD. $p$ values calculated using one-tailed unpaired t-test. ns, not significant ($p > 0.05$). Source data are provided as a Source Data file.

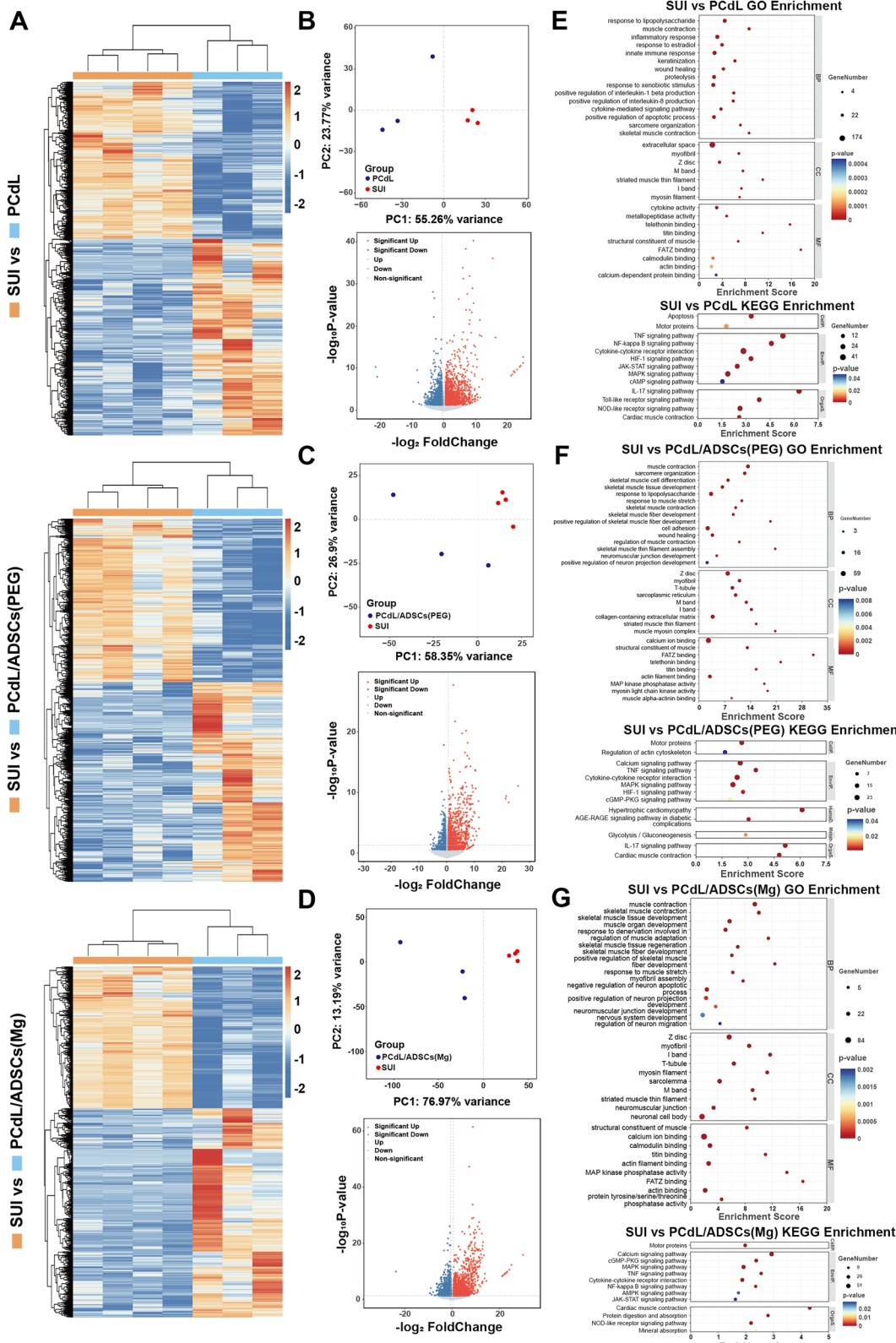

**Fig. 8 | The difference biology in urethral sphincter between the treatment group and the SUI group with different injection systems was explored by RNA-seq. A** Heat maps of urethral sphincter tissues between SUI group and either group among PCdL, PCdL/ADSCs (PEG) and PCdL/ADSCs (Mg); both of which are used to evaluate the whole gene difference. PCA analysis and differential gene volcano mapping between SUI group and either group among (**B**) PCdL, **C** PCdL/ADSCs (PEG) and (**D**) PCdL/ADSCs (Mg). Up-regulated pathways in GO and KEGG enrichment analysis after mRNA sequencing between SUI and either group among (**E**) PCdL, (**F**) PCdL/ADSCs (PEG) and (**G**) PCdL/ADSCs (Mg).

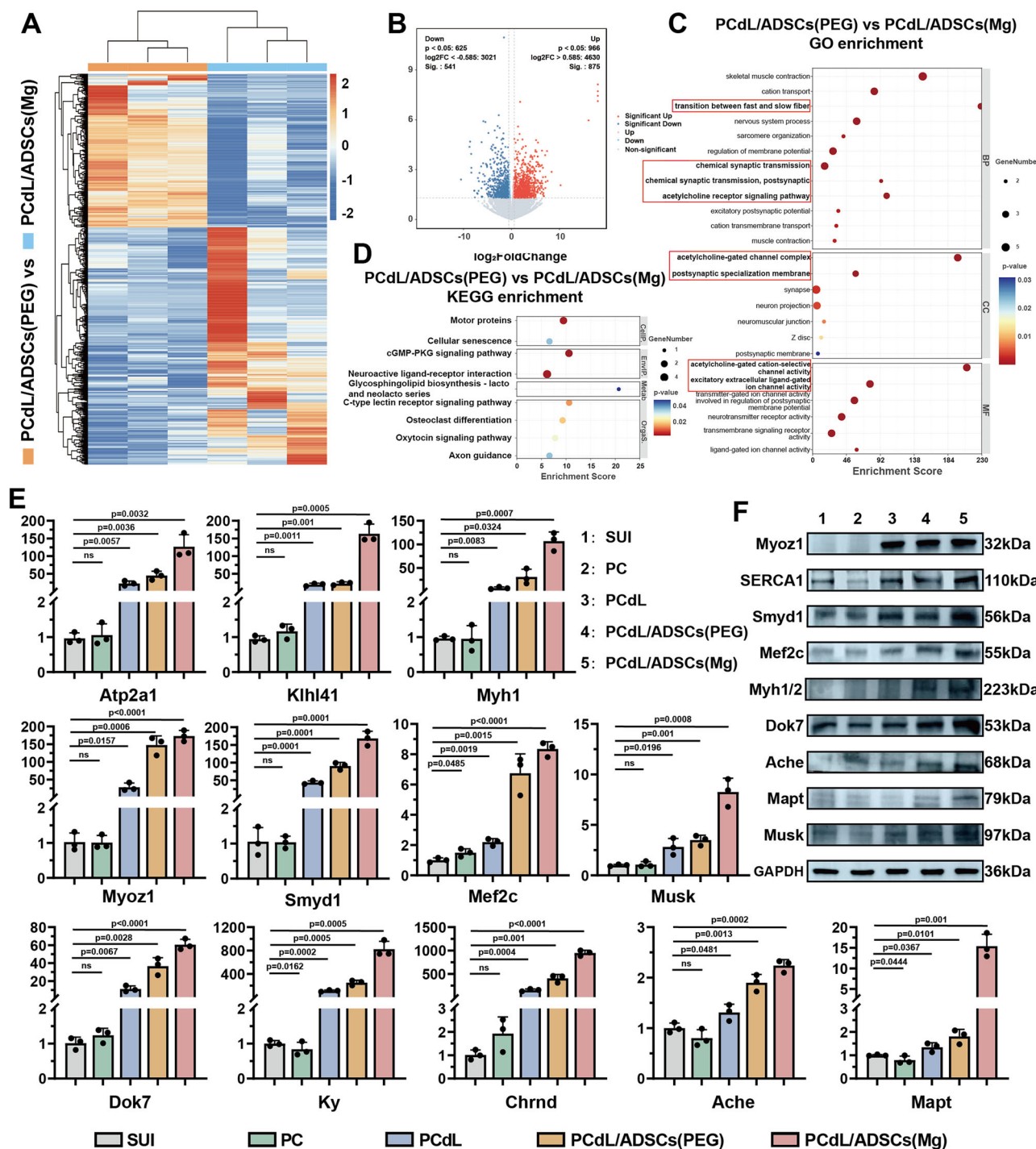

**Fig. 9 | Molecular biological mechanisms of NMJ stabilization and muscle contraction achieved by the PCdL/ADSCs (Mg) injection system. A** Heat maps of urethral sphincter tissues between PCdL/ADSCs (PEG) and PCdL/ADSCs (Mg) group. **B** Differential gene volcano mapping between PCdL/ADSCs (PEG) and PCdL/ADSCs (Mg) group. Up-regulated pathways in (**C**) GO and (**D**) KEGG enrichment analysis after RNA-seq between PCdL/ADSCs (PEG) and PCdL/ADSCs (Mg) group. **E** Expression of genes related to Muscle contraction and regeneration as well as neurogenesis and stabilization of NMJ function of SUI, PC, PCdL, PCdL/ADSCs (PEG) and PCdL/ADSCs (Mg) group via qRT-PCR tests. **F** Representative WB images of urethral sphincters after injection different system for 7 days. Data are expressed as the mean ± SD (*n* = 3 biologically independent samples). All error bars represent SD. *p* values calculated using one-tailed unpaired t-test. ns, not significant (*p* > 0.05). Source data are provided as a Source Data file.

component analysis revealed marked upregulation of neuromuscular junction, while KEGG pathway analysis highlighted enhanced activity in motor protein dynamics and the calcium signaling pathway—both pivotal in synaptic transmission and neuromuscular coordination. Finally, gene set enrichment analysis (GSEA) corroborated that both PCdL/ADSC(PEG) and PCdL/ADSC(Mg)

significantly upregulated key signatures associated with muscle contraction and muscle regeneration (Supplementary Fig. 22), providing robust transcriptional evidence for the multilayered regenerative capacity of the engineered hydrogel system.

To explore the unique neurogenic potential of the PCdL/ADSC(Mg) injectable system and assess the in vivo biological effects of

ZIF-8/PEG200@Mg-mediated ADSC programming, we performed differential gene expression analysis between the PCdL/ADSC(Mg) and PCdL/ADSC(PEG) groups (Fig. 9A). The volcano plot (Fig. 9B) revealed 875 significantly upregulated genes, defined by a fold change ≥ 2 and $P < 0.05$, which were included in downstream enrichment analyses. GO enrichment in the *Biological Process* category (Fig. 9C) highlighted strong enrichment in pathways related to neural transmission, including the acetylcholine receptor signaling pathway and chemical synaptic transmission, postsynaptic. These processes are pivotal for synaptic transmission and neuromuscular communication. In particular, acetylcholine signaling at the NMJ plays a central role in mediating muscle excitation and contraction[45]. Postsynaptic chemical synaptic transmission represents the cascade through which neurotransmitters activate receptors, crucial for neuromuscular coordination and plasticity[46]. Interestingly, enrichment of transition between fast and slow fiber suggests potential remodeling of muscle fiber phenotypes, indicative of myofiber reprogramming under regenerative conditions[47]. GO analysis in Cellular Component and Molecular Function categories also revealed significant upregulation of genes associated with acetylcholine-gated channel complex and cation-selective channel activity, implying enhanced functionality of postsynaptic ion channels and synaptic responsiveness[48]. Further KEGG pathway analysis indicated significant upregulation of the Neuroactive ligand-receptor interaction pathway (Fig. 9D), which encompasses a wide array of neurotransmitter ligands and receptors. This suggests that ZIF-8/PEG200@Mg programmed ADSCs may modulate ligand-receptor networks to amplify and fine-tune neural signaling[49]. Taken together, these results suggest a mechanistic model in which the PCdL/ADSC(Mg) system activates calcium ion channels, thereby promoting intracellular calcium influx. This elevation in $Ca^{2+}$ levels trigger synaptic vesicle fusion with the presynaptic membrane, facilitating the release of acetylcholine (ACh) into the synaptic cleft. The binding of ACh to postsynaptic nicotinic acetylcholine receptors (nAChRs) initiates sodium influx, generates a localized action potential, and induces contraction of the urethral sphincter to restore continence and muscle tone[50]. To mechanistically validate the effects of this injectable system on myogenesis and neurogenesis, we performed qRT-PCR (Fig. 9E) and WB (Fig. 9F) to monitor the expression dynamics of key regulatory genes. For muscle contraction and regeneration, genes including *Atp2a1* (encoding SERCA1, responsible for sarcoplasmic $Ca^{2+}$ reuptake), *Klhl41* (regulates protein degradation and muscle fiber homeostasis[51]), *Smyd1* and *Myoz1* (involved in muscle differentiation and remodeling), as well as *Myh1* and *Mef2c* (critical for myofiber lineage specification[52]), were significantly upregulated. For neurogenesis and NMJ stabilization, we observed increased expression of *Musk* and *Dok7*, key structural regulators of NMJ assembly, as well as *Ky* (involved in calcium sensitivity and nerve-muscle interface), *Chrnd* and *Ache* (modulators of AChR clustering and degradation[52]), and *Mapt* (a microtubule-associated protein essential for axonal stability and synaptic plasticity).

Interestingly, recent studies have identified a strong association between mutations in *Dok7* and congenital myasthenic (CM) syndromes[53], a group of neuromuscular disorders characterized by impaired neuromuscular transmission. *Dok7* encodes an adaptor protein essential for the maintenance and function of NMJs[54]. Loss-of-function mutations in *Dok7* result in defective NMJ formation, leading to early-onset muscle weakness and, in severe cases, perinatal lethality. Mechanistically, *Dok7* functions as an adaptor protein that anchors to muscle, skeletal receptor tyrosine-protein kinase at its N-terminal domain to stabilize MUSK phosphorylation[55]. As a receptor tyrosine kinase, MUSK serves as a key regulator of synaptic differentiation and orchestrates presynaptic specialization[56,57]. Loss-of-function mutations in *Dok7* cause severe defects in MUSK phosphorylation and activation, thereby preventing the recruitment of essential NMJ-associated proteins, such as adapter molecule CRK, and ultimately leading to

impaired synaptic integrity and muscle weakness[58]. In this study, we observed that the PCdL/ADSC(Mg) injectable system significantly upregulated both Dok7 and Musk expression in the urethral sphincter tissue, which likely contributed to enhanced NMJ stability and function. This molecular activation was corroborated at both the mRNA and protein levels, suggesting that the PCdL/ADSC(Mg) platform exerts a dual regenerative effect by promoting muscle regeneration and supporting neuromuscular reinnervation. These findings offer mechanistic insight into the therapeutic efficacy of this system in SUI repair and highlight its potential for restoring neuromuscular functionality through targeted molecular modulation.

## Discussions

SUI is one of the most prevalent conditions in urological surgery and predominantly affects middle-aged and older women. The pathogenesis of SUI is primarily attributed to hormonal fluctuations, muscle and nerve atrophy, and physical insults such as childbirth-induced trauma. These lead to urethral sphincter dysfunction that is characterized by muscle fiber damage, neuromuscular atrophy, and impaired urethral closure mechanisms[59,60]. The clinical management of SUI varies depending on disease severity and different therapeutic approaches are employed[61,62]. Mid-urethral sling (MUS) surgery is a gold standard for SUI treatment with 80–90% short-term success, yet mesh-related complications like erosion and chronic pain are common[63]. Urethral bulking agents (e.g., Bulk amid, collagen, autologous fat) offer temporary improvement but have high re-injection rates and limited durability[64,65]. Importantly, these therapies primarily focus on rebuilding urethral support to improve urinary function but offer limited benefits in terms of addressing the fundamental pathological changes (namely, atrophic sphincter muscles and nerves). In recent years, advances in tissue engineering strategies have markedly improved periurethral tissue augmentation and urethral sphincter function restoration, thus expanding the therapeutic arsenal against SUI. However, studies that specifically target nerve regeneration and neurogenic repair within the urethral sphincter microenvironment remain limited[66]. Notably, given the critical role of neuromuscular interactions in sphincter integrity, the long-term efficacy of current SUI treatments remains suboptimal because of a lack of neural support for sphincter muscle function. Achieving simultaneous urethral sphincter regeneration and neurogenic repair should therefore be regarded as a key research priority for the development of next-generation SUI therapies that are aimed at both treatment and recurrence prevention.

The urethral sphincter comprises a circular muscular structure with internal and external components. Anatomically, from the innermost to the outermost layers, it includes the urethral mucosa and submucosa, smooth muscle (internal sphincter), striated muscle (external sphincter), periurethral muscles, and supportive connective tissues. The internal sphincter is controlled by the sympathetic nervous system and functions via involuntary contractions to maintain urethral closure. By contrast, the external sphincter is innervated by the pudendal nerve and serves as the key muscle for the voluntary control of micturition[67,68]. Given the anatomical complexity of the urethral sphincter and its diverse functions, it remains challenging to achieve the coordinated regeneration of both internal and external sphincter muscles along with local nerve repair and neurogenesis. Nonetheless, an integrated approach is essential for restoring sphincter function and providing long-term therapeutic benefits for patients with SUI.

To address the longstanding challenge of simultaneously promoting muscle regeneration and neural repair in the treatment of SUI, we report a pioneering injectable regenerative platform integrating MOF-programmed stem cells with a thermoresponsive hydrogel matrix. This system not only provides immediate mechanical support to the compromised urethral sphincter but also delivers endogenous

chemical and biological cues that coordinately activate myogenic, angiogenic, and neurogenic repair pathways. By targeting the multi-faceted pathophysiology of SUI, this strategy aims to achieve more complete functional restoration of the sphincter and reduce the risk of recurrence. The thermoresponsive PNIPAm-COOH hydrogel used in this study exhibits precise sol–gel transition behavior. Upon reaching physiological temperature (lower Critical Solution Temperature ≈ 32 °C), the PNIPAm backbone undergoes a sharp phase separation, enabling the injectable precursor solution to rapidly convert into a stable three-dimensional network in situ after administration. This facilitates localized shaping and mechanical reinforcement of the urethral sphincter region. To enhance mechanical integrity, leucine was covalently grafted onto the hydrogel via EDC/NHS-mediated amide bond formation, thereby increasing the crosslinking density of the matrix. This dual modification preserved the thermal responsiveness while significantly improving tensile and compressive moduli, providing sufficient stiffness to withstand repeated increases in abdominal pressure and better support sphincter regeneration. Notably, leucine was stably conjugated to the PNIPAm-C backbone rather than simply blended, leveraging its intrinsic ability to activate the mTOR signaling pathway. As a result, the hydrogel system markedly upregulated myogenic regulatory genes such as *Myoz1* and *Smyd1*, enhancing myogenic differentiation and simultaneously suppressing protein catabolism—demonstrating superior regulatory efficiency over physical mixtures. Previous studies have demonstrated that ADSCs, enriched with a variety of trophic factors, can accelerate muscle regeneration and restore urethral sphincter function. However, their capacity to promote neuronal repair and stabilize NMJs remains limited. To address this shortcoming, we innovatively employed ZIF-8/PEG200@Mg MOF NPs for pre-programming of ADSCs. ZIF-8 nanoparticles, known for their high biocompatibility, structural stability, and tunable surface properties, serve as an ideal nano-carrier platform[69]. $Mg^{2+}$, as essential physiological cofactors, play crucial roles in synaptic signal transmission, NMJ stabilization, axonal outgrowth, and endothelial cell-mediated angiogenesis. To leverage these effects, we encapsulated $Mg^{2+}$ within the ZIF-8 framework and modulated PEG200 surface chemistry and porosity to achieve sustained ion release. This strategy maintains a local $Mg^{2+}$ concentration within an optimal therapeutic window for both neuroregeneration and vasculature remodeling. By allowing ADSCs to internalize ZIF-8/PEG200@Mg nanoparticles prior to hydrogel encapsulation, we endowed the cells with a dual programming effect—pro-neurogenic and pro-angiogenic—while avoiding the risks associated with uncontrolled burst release of free nanoparticles. This cell-internalized delivery approach significantly enhanced the therapeutic efficacy of MOF-based nanomaterials. Synergistic effects were validated in both in vitro and in vivo models. ADSCs programmed with ZIF-8/PEG200@Mg showed upregulation of neurogenic markers such as Dok7 and Musk. In a co-culture model with HUVECs, these ADSCs also promoted robust expression of angiogenic genes. In vivo, SUI rats treated with the composite injectable exhibited substantial improvements in LPP at both 4- and 8-weeks post-injection, accompanied by restored voiding function. Histological and immunofluorescent analyses revealed extensive regeneration of both striated and smooth muscle fibers, augmented neovascularization, and reestablishment of NMJ architecture. Intriguingly, transcriptomic and protein-level analyses (RNA-seq, qRT-PCR, and WB) confirmed activation of NMJ-related pathways. As the functional interface between motor neurons and skeletal muscle, NMJs are essential for neuromuscular signal transmission and maintenance of muscle tone. Their degeneration can lead to denervation and atrophy[70,71]. Importantly, ADSCs programmed with $Mg^{2+}$-releasing MOFs significantly upregulated key NMJ stabilization genes such as *Ky*, *Dok7*, and *Musk*. This is particularly notable, mutations in *Dok7* may cause abnormal phosphorylation and activation of MUSK proteins, leading to NMJ dysfunction thereby causing muscle

weakness and muscle atrophy[72,73]. Upregulation of *Dok7* and Musk thus provides a robust molecular basis for restoring sphincter function via NMJ stabilization and regeneration.

In summary, this study pioneers a dual-function regenerative strategy by integrating MOF-induced neurogenic programming of stem cells with a thermoresponsive hydrogel-based delivery platform. This injectable system not only provides mechanical support to the injured urethral sphincter but also actively promotes regeneration of muscle, vasculature, and neural components. Unlike current clinical interventions that primarily offer passive structural reinforcement—such as bulking agents or sling implants—this approach directly addresses the multifactorial degeneration underlying SUI pathophysiology. It overcomes the limitations of conventional bulking agents that often require repeated administration, and it avoids the surgical complications commonly associated with synthetic slings. Importantly, the delivery system developed herein holds promise for large-scale translation due to its economic and scalable manufacturing. The PNIPAM-COOH hydrogel and leucine components are characterized by low production costs and high batch-to-batch reproducibility. Moreover, ADSCs can be readily harvested in large quantities with minimal invasiveness, and they exhibit robust proliferative capacity and phenotypic stability in vitro. Taken together, the overall low synthesis cost and high manufacturing feasibility make this injectable platform a compelling candidate for future clinical development and industrial-scale application. Despite its promising therapeutic potential, our study has certain limitations. In particular, the mechanical performance of the periurethral tissues after implantation has not been quantitatively assessed due to the anatomical limitations of the rat model, and the long-term in vivo stability and degradation behavior of the hydrogel system warrant further investigation in extended preclinical studies[74]. What's more, Future clinical translation faces several key challenges, including Food and Drug Administration's (FDA) combination product approval process. Critical requirements encompass ensuring material sterility, demonstrating long-term biocompatibility in humans, and conducting comprehensive toxicological assessments. Furthermore, standardization of ADSCs sourcing and programming, including tumorigenicity, migration potential, and batch-to-batch consistency, must be rigorously validated to meet clinical-grade standards[75]. In addition, although the hydrogel materials demonstrated favorable biocompatibility and minimal cytotoxicity in both in vitro and in vivo animal models, their potential to elicit long-term immune responses upon implantation in humans remains to be rigorously assessed. While the decellularization process effectively removes immunogenic components such as residual DNA from the dECM, comprehensive immunological profiling is still necessary to ensure clinical safety and minimize potential adverse responses. Notably, the long-term in vivo stability of the hydrogel plays a pivotal role in determining its therapeutic durability. Standard in vitro degradation assays may not faithfully replicate the complex in vivo degradation kinetics. Therefore, longitudinal preclinical studies—extending over 6 to 12 months in large animal models—will be essential to monitor urethral function, local tissue remodeling, and sustained muscle and neural regeneration, as well as to evaluate chronic toxicity and mechanical persistence. These data will provide critical insights into the translational potential and in vivo durability of the injectable platform. Lastly, SUI is a multifactorial condition with substantial inter-patient heterogeneity, including variations in urethral mobility, sphincter function, and age-related tissue degeneration. These differences can profoundly affect the therapeutic response to biomaterial-based interventions. Hence, for successful clinical translation, future therapeutic strategies employing this injectable system should be tailored to the patient's underlying etiology and physiological characteristics to

optimize efficacy and improve patient satisfaction. Nevertheless, this therapeutic strategy that integrates muscle regeneration and neurogenesis represents a promising direction for future clinical advancements in SUI treatment.

# Methods

## Ethical statement

All animals were acquired from the Laboratory Animal Center of Shanghai Sixth People's Hospital with specific pathogen-free (SPF) status. The rats used in this study belong to the outbred Sprague-Dawley (SD) strain with no genetic modifications. All animal experiments were approved by the Animal Welfare Ethics Committee of Shanghai Sixth People's Hospital under the approval number (2025-0004) and were performed in accordance with relevant guidelines.

## Synthesis of PNIPAm-C polymer

The chain transfer agent 4-(benzenecarbonothioylsulfanyl)-4-cyanopentanoic acid (CPAD), the initiator 2,2′-Azobis(2-methylpropionitrile), the monomer N-isopropylacrylamide (NIPAm) and the tert-butyl acrylate (t-BA) were added to the solution 1,4-Dioxane in a molar ratio of 1:0.5:30:5, and then the reaction was carried out in an oil bath at 70 degrees Celsius for 12 h to remove oxygen by degassing. After the reaction, the product was precipitated with anhydrous ether and dried under vacuum. Subsequently, 1.0 g of the above copolymer was dissolved in dichloromethane and 0.05 mol of trifluoroacetic acid was added to the solution, and the reaction was stirred at room temperature for 48 h. The reaction solution was then fully evaporated to completely remove the dichloromethane and trifluoroacetic acid, and the final PNIPAm-C polymer was obtained.

## Harvest of ADSCs sheet dECM

Briefly, ADSCs were extracted from SD rats (female, 4-week-old, 100-120 g) fresh adipose tissue in low sugar DMEM (Dakewe, 66016111) medium containing 10% FBS (Umedium He Fei, China, 3023 A), 1% penicillin/streptomycin (NEST Biotechnology Co. Ltd. 211092). Once the cells have been passaged to the third generation and are proliferating robustly, the culture conditions are switched to high sugar DMEM (Gibco, 10564011) supplemented with 15% FBS (Gibco, 10099141 C), 20 mg/mL vitamin C (Aladdin, A103539), and 0.5% penicillin/streptomycin. During the cultivation of cell sheets, medium changes should be performed gently to avoid curling or detachment of the sheets due to temperature fluctuations. To minimize inter-animal and inter-batch variability, three independent batches of ADSCs were isolated and expanded, each from three individual female SD rats ($n = 9$ in total). Prior to MOF programming and injection therapy, pooled ADSCs from each batch were thoroughly mixed and then equally distributed across experimental conditions. The obtained ADSCs (P3) membrane sheets were immersed in decellularization solution for decellularization for 72 h. Finally, the decellularized samples were lyophilized and cryogenically ground into microparticles using a grinder, and then sterilized by C060 γ-irradiation.

## Rheology analysis of hydrogel

The process of temperature-sensitive hydrogel gelation was assessed by monitoring the rheological properties of hydrogels using a temperature scanning method (HAAKE RheoStress 6000, Thermo Scientific, USA). The tests were performed using a 10 mm diameter fixture. The temperature was first cooled down to 10°C and subsequently increased at a rate of 1°C/min. The constant strain was 1%, the frequency 1 rad/s, and the temperature range was 10-50 °C. The storage modulus (G′) and loss modulus (G″) were recorded as a function of temperature.

## Degradation ratio assay in vitro

The degradation characteristics of PNIPAm-C and PNIPAm-C/Leu/dECM hydrogels were determined. At the beginning of the experiment, hydrogels with the same quality were selected, three samples in each group. The hydrogels were individually placed in 5 mL of PBS buffer and degraded at 37 °C and 100 rpm in an oscillator. At the end of the oscillation, the samples were collected and lyophilized, and then their weights were measured separately. The degradation rate was calculated according to the equation:

$$D = \frac{(Wo - Wt)}{Wo} \times 100\% \qquad (1)$$

Note: Wt. is the mass of remained hydrogel after 1, 3, 5, 7, 9, 11 and 13 days of degradation and W0 is the initial mass of hydrogel.

## Evaluation before and after decellularization

The DNA content of ADSCs sheets before and after decellularization was estimated by using a PicoGreen DNA kit (Invitrogen, USA). The concentrations of VEGF (U96-1640E), PDGF-BB (U96-2901E), EGF (U96-2054E) and HGF (U96-2401E) in the lysates of the ADSCs sheets extracts before and after decellularization were determined using an enzyme-linked immunosorbent assay (ELISA) kit [YOBIBIBIO Biotech, Shanghai, China] according to the manufacturer's instructions.

## Synthesis of ZIF-8, ZIF-8/PEG200 and ZIF-8/PEG200@Mg NPs

Take 1.098 g of zinc nitrate hexahydrate (Zn (NO3)2·6H2O) and 36 ml of methanol, put them in conical flask 1 and stir for 10 min. Take 2.443 g of 2-methylimidazole (C4H6N2) and 60 ml of methanol, put them in conical flask 2 and stir for 10 min. The solution from conical flask 1 was slowly added to conical flask 2 and continued to be stirred for 1 h. After standing for 24 h, centrifuge and wash with methanol 3 times to get ZIF-8. After that, 60 mL of PEG-200 was added and stirred for 1 h. The solution was allowed to stand for 24 h, centrifuged and washed with methanol for 3 times to obtain ZIF-8/PEG200. 2 g of anhydrous magnesium chloride was added to 20 mL of water and stirred for 10 min. 0.5 g of the prepared ZIF-8/PEG200 powder was added and stirred for 1 h. The solution was centrifuged to obtain ZIF-8/PEG200@Mg.

## Characterization of MOF NPs

The FTIR spectra of hydrogels were studied under IR irradiation at different wavelengths using a FOLI10 Conventional FTIR Spectrometer (Yinsa optical, INSA, Shanghai, China) at room temperature. We used DLS (ZEN3600, Malvern) to examine the different particle sizes of the three MOF NPs, and the ζ-potential was measured at room temperature using a Malvern Zetasizer nano-ZS90. Physical analysis of the crystal structure was performed by X-ray diffraction (XRD) (X'Pert PRO MPD). Surface morphology and elemental mapping analysis of the MOF NPs were observed using a transmission electron microscope (TEM) (FEI Tecnai F20) with an energy dispersive spectrometer (EDS) accessory. We detected the chemical elemental compositions of the three MOFs using an ESCALAB QXi *XPS* spectrometer (Thermo Fisher Scientific, USA). The *XPS* spectra were analyzed by a *XPS* PEAK software to conduct peak separation. We measured the pore size distribution of the three MOFs by applying the N2 adsorption isotherm at 77 K through nonlocal density functional theory.

## Biocompatibility and biological function assay

Cell Counting Kit 8 (CCK 8) (New Cell & Molecular Biotech, C6005) was used to assess the biosafety of different injection systems and the ability to promote cell proliferation and growth. Briefly, ADSCs were co-cultured with different hydrogel systems for 1, 4, and 7 days in low sugar DMEM (Gibco, 11885084) supplemented with 10% FBS (Cyagen,

FBSSR-01021) and 1% penicillin/streptomycin (Gibco, 15140122). The absorbance of the supernatant at 450 nm was analyzed using a microplate reader and optical density values were recorded. Cell survival of ADSCs after 3 days of co-culture with different hydrogels was assessed in vitro using a live/dead cell staining kit (Elabscience BiotechnologyCo., Ltd., E-CK-A354). Petri dishes (CellPro Biotechnology, 803100B) were rinsed three times with PBS and then treated with PBS solution containing propidium iodide (PI) and Calcein-Acetoxymethyl Ester (AM) at room temperature, followed by incubation in the dark for 30 min, and finally photographed with a fluorescence microscope. The percentage of live and dead cells was calculated after counting with ImageJ software.

Monolayers were manually scratched with a 200 μL specialized pipette tip. Subsequently, digital photographs of the wound area were taken with a light microscope (Olympus, Japan) at 0, 12, 24, and 48 hours, respectively. Cell-free areas were measured and analyzed by ImageJ to assess the ability of different hydrogels to promote cell migration.

### Tube-formation experiments in co-culture systems

First, ADSCs were seeded in the upper chamber of a 24-well Transwell and cultured with 10% FBS (JYK-FBS-300, Jin Yuan Kang Biotechnology) for 48 h. After 12 h of starvation treatment with serum-free medium, the medium was replaced with complete medium followed by the addition of a certain amount of MOF NPs to allow the cells to phagocytose for 48 h. Subsequently, Matrigel (Yeasen Biotechnology (Shanghai) Co., Ltd., 40186ES08) was added to the lower chamber and allowed to stand at 37°C for 1 h. HUVECs were then placed into Matrigel-coated wells and divided into groups according to the treatments received. After 6 hours of incubation at 37°C in an incubator with a carbon dioxide concentration of 5%, three random visual zones were collected and analyzed for tube length and total branch points using ImageJ software. Analysis was performed using ImageJ software.

### Evaluation of the effects of different injection systems for the treatment of SUI in rats

All experiments were conducted using 4-week-old female SD rats. The use of female animals was based on the clinical relevance of SUI, which predominantly affects women. SD rats were divided into 5 groups: SUI, PC, PCdL, PCdL/ADSCs (PEG) and PCdL/ADSCs (Mg) groups (three rats per group), and the rat SUI model was constructed by vaginal balloon dilatation and removal of bilateral ovaries. Briefly, rats were anaesthetized by exposure to 20 % ketamine, and both right and left ovaries were then removed. A 12 F catheter (Bard, Kulim, Malaysia) was inserted into the vagina, followed by injection of 2.5 mL of PBS into the air sac to compress the vaginal wall. The F12 catheter was leveled and attached with 120 mg of weight for 4 h before removal. Saline, PC, PCdL, PCdL/ADSCs (PEG) and PCdL/ADSCs (Mg) hydrogels were injected around the bladder neck urethral sphincter at 4 W after modeling, and the rats were subjected to LPP assay at 4 and 8 weeks after the injection treatment, respectively. Briefly, a polyethylene-90 catheter was inserted through the bladder dome and the bladder was continuously filled with sterile saline while the increase in intravesical pressure was recorded with a pressure transducer (ADInstruments, Castle Hill, New South Wales, Australia) until leakage occurred. The pump flow rate was 6 ml/h. Twenty cycles were run and three stable cycles were selected for graphing. The pressure at which leakage occurs was defined as the LPP ($n = 10$). After testing, rats were anesthetized and killed by neck dissection, the urethra and bladder tissues were carefully separated, and the urethral sphincter tissues of the bladder neck were immersed in tissue fixation solution (RCF-02, Huilanbio BiologicalTechnology, Shanghai, China) and fixed for 24 hours. After dehydration using graded ethanol, these tissues were embedded in paraffin and sections (4 μm) were examined histologically by HE staining and Masson staining.

### Immunofluorescence analysis

The expression of urothelial Desmin, α-SMA, Synaptophysin, TUBB3, α-BTX, SV2A and VEGF was analyzed by immunofluorescence staining in each group (three rats per group). Briefly, slides were incubated overnight at 4°C with the corresponding primary antibody for each index and then placed in a wet box containing a small amount of water. The slides are washed with PBS in a shaker setup, followed by covering the target tissue with the secondary antibody (properly conjugated to the primary antibody) and incubating for 50 minutes at room temperature in the dark. DAPI staining: The slides are washed with PBS in a shaker setup, followed by incubation with DAPI solution for 10 minutes at room temperature in the dark. Afterwards, the slides were examined and captured by fluorescence microscope, the excitation wavelength of DAPI was 330-380 nm, the emission wavelength was 420 nm, and the color was blue; the excitation wavelength of FITC was 465-495 nm, the emission wavelength was 515-555 nm, and the color was green; the excitation wavelength of CY3 was 510-560 nm, the emission wavelength was 590 nm; semi-quantitative analysis was carried out using ImageJ software. ImageJ software for semi-quantitative analysis.

### qRT-PCR analysis

Gene expression levels of cells extracted from bladder neck urethral sphincter 7 days after injection of different hydrogels (three rats per group) were analyzed by qRT-PCR. qRT-PCR was performed on the detection system using SYBR Green Pro Taq HS qPCR Kit II (Rox Plus) (ACCURATE BIOTECHNOLOGY (HUNAN) CO., LTD, ChangSha, China (AG11719)). The forward and reverse primer sequences of *β-actin, Acta2, Atxn7, Claps2, Fez1, Fgfbp3, Kirrel3, Wasf2, Nptxr, Ntn1, Itga2, Plxnb1, Shank3, Atp2a1, Klhl41, Myh1, Myoz1, Smyd1, Mef2c, Musk, Dok7, Ky, Chrnd, AChE* and *Mapt* are shown in Supplementary Table 1 of the Supporting Information. mRNA expression levels were quantified by using β-actin as an internal control and data were determined based on the cycle thresholding method as $R = 2^{-\Delta\Delta CT}$ ($n = 3$).

### Western blotting analysis

New tissues were harvested 7 days after the different treatments (three rats per group), protease and phosphatase inhibitors were added, and the tissues were lysed in RIPA lysis buffer and ground. Cell lysates were allowed to stand on ice for 1 hour and then centrifuged to collect the supernatant. Each lane was loaded with a total of 15 μg of protein and electrophoresed using an 8-12% SDS-PAGE gel. The target proteins on the SDS-PAGE gels were transferred to polyvinylidene difluoride membranes (PVDF; 0.45 μm) and then closed with Rapid Protein Free Closure Solution (BR0051-01, Boyi Biotech, China) for 1 h at room temperature. The PVDF membranes were treated with primary antibodies against β-actin, Acta2, Atxn7, Fez1, Claps2, Shank3, Myoz1, SERCA1, Smyd1, Mef2c, Myh1/2, Dok7, Ache, Musk and Mapt (1:1000), respectively, at 4°C overnight. The membranes were then rinsed three times with TBST and incubated with the corresponding secondary antibodies for 1 hour, followed by contrast development for photographs. Antibody sources and Catalog Number are detailed in Supplementary Table 2.

### Statistical analysis

All data are presented as the means ± SDs. All error bars represent SD. P values calculated using one-tailed unpaired t-test with $P < 0.05$ considered statistically significant.

### Reporting summary

Further information on research design is available in the Nature Portfolio Reporting Summary linked to this article.

## Data availability

Source data are provided with this paper. The authors declare that all data supporting of results in this study are available within the paper

and its Supplementary Information, or from the corresponding authors upon request. The raw RNA-seq data generated in this study have been deposited in the Genome Sequence Archive (GSA) database under accession code CRA027005 and CRA028370 (https://ngdc.cncb.ac.cn/gsa). The processed expression data used in this study are provided in the Source Data file. All unique materials used are readily available from the authors or from standard commercial sources. Source data are provided with this paper.

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

## Acknowledgements

The authors are grateful to the ceshihui (www.ceshihui.cn) for providing invaluable assistance with NMR, DSC, and rheological properties testing; BioRender.com, which was used to provide the icons of the illustrations. This research was financially supported by the National Natural Science Foundation of China (No. 82100714 to Y.W., No. 82370684 to Q.F. and 82170694 to Q.F.), the Interdisciplinary Program of Shanghai Jiao Tong University (No. YG2021QN102 to Y.W. and No. YG2022ZD020 to Q.F.), the 2022 Shanghai Leading Talent Training Program (Grant No. 026 to Q.F.), the Natural Science Foundation of Shanghai (No.20ZR1442100 to Q.F.), the Program of Shanghai Sixth People's Hospital (Grant No. ynts202004 to Y.W.).

## Author contributions

W.F., X.D., and R.Y. contributed equally to this work. Y.W., Q.F., and G.G. conceptualized the study. W.F., and X.D. developed the methodology. R.Y., M.L., and M.Y. conducted the investigation. W.F., R.Y., and Y.J. were responsible for visualization. Y.W., Q.F., and G.G. supervised the study. W.F. and R.Y. wrote the original draft. Y.W. and Q.F. reviewed and edited the manuscript.

## Competing interests

The authors declare no competing interests.
