## [Transparent Peer Review file · Nature Communications]

Injectable therapeutic system incorporating neurogenesis-programmed stem cells concomitantly promoting muscle regeneration treats stress urinary incontinence

Corresponding Author: Professor Ying Wang

Version 0:

Reviewer comments:

Reviewer #1

(Remarks to the Author)

This manuscript reports early animal studies of a novel hydrogel injectable system for use in surgery for stress incontinence of urine. This is a timely piece of research due to the undertreatment of women with SUI in some jurisdictions due to the pause on the use of polypropylene due to erosion / chronic pain. The authors report that the injectable system promotes urethral sphincter muscle regeneration and nerve regeneration. As such, this represents an addition to the literature and is therefore worthy of publication. The authors are to be commended on what is a large body of original work with multiple stages which clearly required a lot of time and dedication to complete. Nonetheless, the manuscript suffers from limitations that preclude its publication.

Qs for authors;

Why did you choose ADSCs as your source of stem cells? Explain rationale

Is ZIF-8 currently in use in humans in any other implantable devices?

Authors reports increased presence of NMJ, nerve fibres, and regeneration of smooth/striated muscle in rats following administration of the injectable system. This is the key difference that would potentially make this surgical procedure superior to bulking agents / autologous slings. LPP measurements at 4 and 8 weeks showed improved sphincter function, what long term duration of efficacy do the authors expect to see with this injectable and how will they assess this in future studies ? Will the injectable undergo degradation over time such as with urethral bulking agents and require repeat procedures? This injectable should be compared against the current standard of care which is urethral bulking agents or autologous sling, was this possible to include in the SUI rats study

What kind of delivery system do the authors expect to use in human trials?
Expected cost?

Do the authors have any concerns about the biocompatibility of this injectable in humans? Live/dead staining of the injectable co cultured with ADSCs showed favourable results.

Regarding the design of the animal model; "4-week-old female rats underwent vaginal balloon dilation combined with bilateral ovariectomy to simulate the primary etiology observed in most clinical SUI patients." What was the rationale for designing the study in this way and are there any limitations in its design. Is 4 weeks long enough post bilateral ovariectomy to simulate post menopausal SUI?

When looking at the elevated LPPs, is urinary retention a concern?

Limitations of the study is not discussed enough in the Discussion section. The authors state in the final paragraph of the Discussion section "several challenges remain for its clinical translation, including large-animal validation and the evaluation of potential immune responses to foreign materials." This is not enough and needs to be expanded upon. The injectable was not compared against the gold standards; bulking agents and autologous slings. Was mechanical texting on explanted pelvic floor tissue performed? Barriers to getting the injectable material approved for human trials? Long term durability not assessed?

Reviewer #2

(Remarks to the Author)

The study looked at the use of modified adipose-derived stem cells (ADSC) for the treatment of stress urinary incontinence in rats. They used an established childbirth injury model as well as leak point pressure testing and cytometry to assess lower urinary tract function. Additionally, they used many standard techniques to demonstrate the ADSC potential in culture. They found the cell-enhanced expression of many regenerative-associated genes when the cells were combined with nanoparticles. They saw a significant increase in function outcomes when the rats were treated with the ADSC with nanoparticles. Additionally, they saw an increase in anatomical outcomes.

Concerns.

1. I believe there is a typo in Figure 5's legend. It states that an N of 3 was used. An N of 3 would be ok for histology, but an N of 8 or more would be needed for the lower urinary tract cytometry. Additionally, are you sure those error bars are standard deviation? Those standard deviations are minimal compared to others published for cystometry (Kuo 2001, Urologia Internationalis). Please clearly state how many animals were used in the functional assessments in the Methods section and the figure legend.
2. More details are needed in the methods section to help readers understand the experiments. Please provide more information about the vaginal distension (VD) and ovariectomy. For the VD, please give the size of the dilators used, the balloon size, the volume it was filled with, and how long it was inflated. These can vary from lab to lab, so it is best to state them clearly. Please justify starting the experiment with 4-week-old rats. Most studies use older rats. Was a separate cohort of rats used for 4 weeks vs 8 weeks?
What was the flow rate for the cytometry? How many cycles did you run for data?
3. Please provide more details about the stem cells. How many times did you expand them? How many animals were used for harvesting? What were the animals' ages and sexes? Did you select for stem cells, and if so, what markers did you use? How many separate batches of cells were harvested?
4. Please provide a table with all the abbreviations and their meaning.
5. For all figures, please ensure you define all abbreviations in the legend. For Figure 2 in panel b, I am assuming a.u. stands for arbitrary unit. Additionally, in the corners of panel c immunofluorescent images, there is an indicator that they should be more prominent in size because they are currently unreadable. If they are not required, remove them.
6. For Figure 3, were replicates for the n of 3? Were three separate batches of cells used, or were they from all the identical batches?
7. In Figure 4, panel F, no Propidium Iodide (PI) stained cells are visible in the image, but your pie charts state that some PI-positive cells should be present. I recommend showing an example image with some PI-positive cells to demonstrate that the PI staining was done correctly.
8. In Figure 5 panel B, please include a time scale for each of the bladder traces or make all the traces to the same time scale and provide that scale in one of the traces.
9. In Figure 8 panel A, the diagram is too small to read. I recommended either making a more simplified version or removing it.
10. Please consider doing some triple staining for NMJs and nerve axons. Many groups do staining to determine the number and percentage of innervated NMJs.

Reviewer #3

(Remarks to the Author)

This manuscript presents a novel injectable therapeutic system combining MOF nanoparticles and ADSCs hydrogel for the treatment of stress urinary incontinence (SUI), addressing critical challenges associated with muscle regeneration, angiogenesis, and neurogenesis. The study is innovative and demonstrates a well-integrated interdisciplinary approach that leverages advanced biomaterials and stem cell engineering to tackle an unmet clinical need. The experimental design is robust, supported by comprehensive in vitro and in vivo analyses, and highlights the synergistic effects of the materials used. However, the paper's clarity and impact are hindered by verbose sections, insufficient emphasis on the translational potential, and underexplored discussions on clinical challenges. By addressing these aspects, this study has the potential to offer significant insights into next-generation regenerative therapies.

1. The abstract is overly verbose and does not clearly emphasize the novelty of the work. The purpose, main findings, and significance of the study are not presented with sufficient clarity.
2. The manuscript suffers from inconsistent transitions between sections. For instance, the introduction of MOF nanoparticles mixes background information and experimental details, hindering coherence.
3. Despite the manuscript's innovative use of MOF nanoparticles and ADSCs hydrogel systems, this synergy is not emphasized enough. The rationale behind the combination is not clearly articulated, particularly regarding how it addresses

the limitations of existing SUI treatments.

4. The descriptions of the PNIPAm hydrogel and MOF nanoparticles are too general. The linkage between their specific properties (e.g., thermosensitivity, crosslink density, magnesium delivery) and their functional advantages in nerve and muscle regeneration is insufficiently discussed.

5. Figures, especially those involving RNA sequencing data (e.g., Figures 7 and 8), include dense information without sufficient explanation or cross-referencing in the text.

6. Although statistical significance is provided for most experiments, some datasets lack detailed quantitative analysis. The functional evaluation of NMJ restoration, for example, would benefit from additional metrics like synaptic density or acetylcholine receptor characterization.

7. The discussion does not delve deeply into the practical challenges of clinical translation, such as potential immune responses, large-scale production, long-term material stability, or variability across patient-specific applications of the treatment system.

Version 1:

Reviewer comments:

Reviewer #1

(Remarks to the Author)

The authors have sufficiently addressed all my concerns from my initial review and it is my opinion that the article is suitable for publication

Reviewer #2

(Remarks to the Author)

Thank you for addressing my comments.

Reviewer #3

(Remarks to the Author)

We are deeply grateful to all reviewers for their careful, insightful, and constructive comments, which have greatly helped us improve the quality of the manuscript. All revisions to the manuscript have been marked up using the “Track Changes” function. The following is a point-by-point explanation of the details of the revisions to the manuscript and responses to the reviewers' comments. The responses are written in blue, and the corrections were made in red texts with the exact location in the “Revised Manuscript with Changes Marked” version indicated at the beginning.

Point-by-point response to reviewer 1:

1. Why did you choose ADSCs as your source of stem cells? Explain rationale

Response: Thanks for your valuable questions. In the field of tissue engineering and regenerative medicine, a broad array of stem cell types—including bone marrow-derived mesenchymal stem cells (BM-MSCs), embryonic stem cells, hematopoietic stem cells, and adipose-derived stem cells (ADSCs)—have been explored for therapeutic applications¹. The choice of an optimal stem cell source largely depends on the targeted tissue type and the desired regenerative outcome. In this study, we selected ADSCs as our cell source, owing to their several practical and biological advantages: they are readily accessible in large quantities via minimally invasive procedures, exhibit low immunogenicity, and are suitable for both autologous and allogeneic transplantation^{2,3}. Importantly, ADSCs maintain robust proliferative capacity and multipotent differentiation potential under in vitro expansion, enabling scalable production⁴. Notably, ADSCs have been consistently shown to promote muscle regeneration, angiogenesis, and tissue remodeling⁵, making them a rational candidate for our injectable therapeutic strategy for stress urinary incontinence (SUI).

(Page 3, line 29-32) Owing to their favorable biological characteristics—such as low immunogenicity, stable proliferative capacity, and multilineage differentiation potential—ADSCs have emerged as a promising cell source widely utilized in tissue engineering and regenerative medicine applications^{3,4}.

2. Is ZIF-8 currently in use in humans in any other implantable devices?

Response: Thanks for your valuable questions. To date, ZIF-8-based nanoparticles have not yet been approved for use in human implantable medical devices. While ZIF-8 exhibits considerable potential in biomedical applications—owing to its high porosity, excellent biocompatibility, and pH-responsive behavior—its clinical translation remains at a preclinical stage⁶. Current investigations are primarily focused on in vitro and animal studies, where ZIF-8 has shown promise in drug delivery, wound healing, and tissue regeneration. For example, injectable ZIF-8-based nanocomposite hydrogels have been engineered to accelerate diabetic wound healing, demonstrating favorable cytocompatibility and angiogenic effects⁷. However, such applications

have not yet progressed into human clinical trials. Despite encouraging results, several challenges remain before ZIF-8 nanoparticles can be safely integrated into human implantable devices. These include limited understanding of their long-term biosafety in vivo, potential toxicity of degradation byproducts, and interactions with complex human tissue microenvironments. Rigorous preclinical investigations and tightly controlled clinical trials are therefore essential to assess their safety, degradability, and therapeutic efficacy. In summary, although ZIF-8 nanoparticles represent a highly promising platform in biomedicine, they have not yet been approved for clinical use in implantable medical devices. Further systematic research is warranted before their widespread translation into human applications can be realized.

3. Authors reports increased presence of NMJ, nerve fibers, and regeneration of smooth/striated muscle in rats following administration of the injectable system. This is the key difference that would potentially make this surgical procedure superior to bulking agents / autologous slings. LPP measurements at 4 and 8 weeks showed improved sphincter function, what long term duration of efficacy do the authors expect to see with this injectable and how will they assess this in future studies? Will the injectable undergo degradation over time such as with urethral bulking agents and require repeat procedures? This injectable should be compared against the current standard of care which is urethral bulking agents or autologous sling, was this possible to include in the SUI rats study?

Response: Due to the relatively low density of hydrolytically labile groups in the PNIPAm-C backbone⁸, the biodegradation of the PCdL injectable system primarily arises from the embedded bioactive peptides within the hydrogel network. In vitro degradation assays revealed that after two weeks of incubation, approximately 80% of the PNIPAm-C hydrogel mass remained, while the PCdL system exhibited ~60% residual mass (Fig. 1J). However, it should be noted that in vitro degradation kinetics do not fully recapitulate the complex enzymatic and cellular environment in vivo. Based on preliminary observations, we estimate the therapeutic window of the injectable matrix to last for at least two months in situ.

Unlike conventional urethral bulking agents that rely solely on passive mechanical support, our system offers a dual-action therapeutic mechanism—combining urethral support with active stimulation of muscle regeneration and neurogenesis. Therefore, even after the hydrogel loses its bulking capacity during the later stages of degradation, its early promotion of tissue remodeling may sustain improved urethral sphincter function without the need for repeat injections.

Nonetheless, the precise duration and robustness of long-term efficacy, as well as safety in vivo, require further systematic validation. Future studies should incorporate longitudinal assessments

including monthly LPP and maximum voiding pressure (MVP) measurements, behavioral metrics such as voiding frequency and leakage episodes, and histological/immunohistochemical analyses (e.g., Myo21, Smyd1 for myogenesis; Ache, Dok7 for NMJ regeneration) to monitor sustained tissue regeneration and functional outcomes over time.

Recently, a variety of urethral bulking agents have been clinically employed for the treatment of SUI, including bovine collagen, polyacrylamide hydrogel (e.g., Bulkamid), and autologous fat. In this study, we selected normal saline rather than a bulking agent as the control group. This decision was informed by prior studies in which conventional bulking agents (collagen), were used as positive controls in SUI treatment models⁹. Those studies demonstrated that while urethral bulking agents significantly increased LPP at 2 weeks post-injection, their efficacy diminished substantially by 4 weeks, with LPP returning to baseline levels. These findings suggest that conventional bulking agents offer only transient functional improvement and lack long-term therapeutic efficacy. Based on this, we did not include collagen or other bulking agents as comparators in the present study. Instead, our designed injectable hydrogel system was evaluated against saline controls to highlight its superior and sustained regenerative capacity, targeting not only urethral support but also muscle and nerve regeneration.

4. What kind of delivery system do the authors expect to use in human trials? Expected cost?

Response: In this study, the PCdL/ADSC(Mg) delivery system not only enhanced angiogenesis and muscle regeneration in the urethral sphincter of SUI model rats but, more importantly, significantly promoted the regeneration of nerve fibers and the expression of genes and pathways associated with NMJ stabilization. These combined effects led to a marked improvement in urinary continence function. Given its promising regenerative performance, we envision future clinical translation of the PCdL/ADSC(Mg) delivery platform. The anticipated clinical cost of this system is expected to remain within a reasonable range, mainly derived from two components: the synthesis of the injectable hydrogel and the preparation, culture, and expansion of ADSCs. The PNIPAM-COOH hydrogel, synthesized via free radical polymerization, utilizes readily available laboratory reagents with relatively low production costs¹⁰. Leucine is produced mainly by microbial fermentation or chemical synthesis. Microbial fermentation is currently the main way of leucine industrial production, using engineered strains of bacteria under specific culture conditions to produce leucine in large quantities, with the advantages of low cost and high yield¹¹. ADSCs, known for their abundance, low immunogenicity, and robust proliferative potential, are well-suited for autologous and allogeneic transplantation. Their ease of harvest and expansion under standard culture conditions further lowers their manufacturing cost, making them an ideal cell source for clinical

applications. Overall, if the PCdL/ADSC(Mg) delivery system is applied to future clinical trials, the total cost is expected to be low and favorable for clinical trials.

5. Do the authors have any concerns about the biocompatibility of this injectable in humans?

Live/dead staining of the injectable co cultured with ADSCs showed favorable results.

Response: The thermosensitive PNIPAM hydrogel used as the injectable matrix has been shown in multiple studies to possess excellent biocompatibility. Cytotoxicity assays using 3T3-L1, HEK293, and A549 cell lines, along with neutral red uptake and MTT assays, indicated no cytotoxic effects¹². Single-cell gel electrophoresis (comet assay) showed no DNA damage, and cells maintained normal morphology and viability on hydrogel surfaces. In our study, ADSCs cultured within the hydrogel maintained high viability and proliferative capacity. These results collectively confirm the short-term biocompatibility of PNIPAM-based hydrogels, suggesting their safety for regenerative applications in human SUI. Nevertheless, comprehensive evaluation of long-term in vivo degradation, immunogenicity, and tissue interaction remains essential.

6. Regarding the design of the animal model; “4-week-old female rats underwent vaginal balloon dilation combined with bilateral ovariectomy to simulate the primary etiology observed in most clinical SUI patients.” What was the rationale for designing the study in this way and are there any limitations in its design. Is 4 weeks long enough post bilateral ovariectomy to simulate post-menopausal SUI?

Response: The animal model employed—vaginal balloon distension (VBD) combined with bilateral ovariectomy (OVX)—was designed to mimic the two primary etiological factors of female SUI: trauma-induced injury to the pelvic floor muscles and nerves, and estrogen deficiency post-menopause. VBD induces transient mechanical stretch injury to the pudendal nerve and external urethral sphincter, mimicking childbirth-related pelvic trauma and resulting in a significant drop in LPP¹³. OVX leads to a sharp decline in estrogen, and by 4 weeks post-surgery, urethral basal pressure and LPP are significantly reduced, simulating postmenopausal degeneration of urethral structure and function¹⁴. Although the model can simulate the etiology of clinical SUI in women to some extent, it still has some limitations. For example, due to anatomical and physiological differences between humans and rats, the anatomical stresses on the venous plexus of the pelvic floor in rats are different from those in humans, and the urethral angle and sphincter response to birth injury and hormonal deficiencies may not be exactly equivalent¹⁵. Moreover, the model emphasizes acute nerve-muscle damage and hormonal loss, without addressing the role of connective tissue or chronic inflammation and tissue remodeling commonly seen in human

patients¹⁵.

Several studies have shown that significant decreases in baseline urethral sphincter pressure and leak point pressure can be observed as early as 3 weeks after bilateral oophorectomy, and that the SUI behaves more consistently at 4-6 weeks postoperatively¹⁴. This can mimic the pathophysiological changes in the human postmenopausal SUI to some extent. The results of these studies have been shown in a number of studies.

(Page 17, line 11-14) Briefly, 4-week-old female rats underwent vaginal balloon dilation combined with bilateral ovariectomy to mimic the two primary etiological factors of female SUI: trauma-induced injury to the pelvic floor muscles and nerves, and estrogen deficiency post-menopause.

7. When looking at the elevated LPPs, is urinary retention a concern?

Response: When the SUI rats significantly increased LPP, it indicated that the urethral resistance was enhanced, which is one of the key mechanisms for preventing urine leakage, but if the urethral resistance exceeds the effective contraction force of the urethral muscle, it may lead to an increase in residual urine, or even urinary retention. In this study, the LPP of normal SD rats was at 40.01±2.28 cm H₂O. In the PCdL/ADSC(Mg) delivery system treatment group, the LPP of the rats was 38.53±2.4 cm H₂O, which was a significant enhancement compared to that of SUI rats at the most, but it did not exceed that of the normal SD rats, and thus the phenomenon of urinary retention was not a concern for the time being. In addition, it has been shown that, combining clinical and animal studies, the target LPP should be controlled within 20-50 cm H₂O to prevent leakage without causing significant residual urine or urinary retention¹⁶.

8. Limitations of the study is not discussed enough in the Discussion section. The authors state in the final paragraph of the Discussion section “several challenges remain for its clinical translation, including large-animal validation and the evaluation of potential immune responses to foreign materials.” This is not enough and needs to be expanded upon. The injectable was not compared against the gold standards; bulking agents and autologous slings. Was mechanical texting on explanted pelvic floor tissue performed? Barriers to getting the injectable material approved for human trials? Long term durability not assessed?

Response: We appreciate your suggestion to enhance the quality of the manuscript.

Mid-urethral sling (MUS) surgery remains one of the clinical gold standards for the treatment of SUI, primarily relying on mechanical suspension to restore sub urethral support. While objective cure rates within 1–5 years are reported to be 80–90%¹⁷, mesh-related complications such as erosion and chronic pain remain non-negligible and can substantially compromise patient quality of life.

Urethral bulking agents—including bovine collagen, polyacrylamide hydrogel (e.g., Bulkamid), and autologous fat—achieve short-term (<2 years) improvement rates of 60–70%, with 5–8 year durability ranging from 50–65%. However, re-injection rates have been reported as high as 0–77%, with a median time to repeat treatment of just 3 months. Such frequent retreatments can increase patient burden and healthcare costs¹⁸. In contrast to these two standard modalities, our injectable system offers several unique advantages. The thermoresponsive hydrogel ensures a liquid state at low temperatures for minimally invasive delivery and undergoes rapid gelation at physiological temperature, enabling precise microenvironmental filling around the external urethral sphincter. The incorporation of leucine activates the mTOR signaling pathway to promote protein synthesis and muscle regeneration, while MOF-programmed ADSCs potentiate neurogenesis and NMJ reconstruction. This dual-function (regeneration + support) strategy may reduce the need for repeat injections and mitigate foreign body-related complications.

Although our histological analysis and LPP measurements confirmed structural and functional improvement of the urethral sphincter, mechanical testing of pelvic tissues post-implantation was not performed due to the anatomical limitations of the rat model. Future studies in large animal models will aim to include biomechanical characterization to evaluate the effect of the injectable system on pelvic tissue integrity and compliance.

From a translational perspective, several regulatory barriers must be addressed before clinical trials can be initiated. The hydrogel represents a combination product consisting of both biomaterial and biologics, which would require a coordinated regulatory pathway, such as that outlined by the FDA’s Office of Combination Products. PNIPAM-based materials must be sterile, exhibit acceptable biocompatibility, and undergo comprehensive toxicological evaluation. It is also necessary to ensure ADSCs sources and programming consistency, tumor-free and migratory assays, and batch-to-batch stability. Long-term, systematic clinical trials will be essential to establish safety and efficacy, with early-phase trials focusing on adverse events and improvements in voiding function¹⁹.

Finally, the long-term durability of this therapeutic strategy remains to be fully established. While our current study is limited to short-term efficacy in rodents, future work will involve longitudinal studies in large animal models over 6–12 months, including repeated assessments of urethral function, local tissue remodeling, and sustained muscle and nerve regeneration, alongside systemic toxicity monitoring²⁰.

(Page 26, line 24-27) Mid-urethral sling (MUS) surgery is a gold standard for SUI treatment with 80–90% short-term success, yet mesh-related complications like erosion and chronic pain are common¹⁷. Urethral bulking agents (e.g., Bulk amid, collagen, autologous fat) offer temporary improvement but have high re-injection rates and limited durability^{18,21}.

(Page 29, line 2-11) Despite its promising therapeutic potential, our study has certain limitations. In particular, the mechanical performance of the periurethral tissues after implantation has not been quantitatively assessed due to the anatomical limitations of the rat model, and the long-term in vivo stability and degradation behavior of the hydrogel system warrant further investigation in extended preclinical studies²⁰. What's more, Future clinical translation faces several key challenges, including Food and Drug Administration's (FDA) combination product approval process. Critical requirements encompass ensuring material sterility, demonstrating long-term biocompatibility in humans, and conducting comprehensive toxicological assessments. Furthermore, standardization of ADSCs sourcing and programming, including tumorigenicity, migration potential, and batch-to-batch consistency, must be rigorously validated to meet clinical-grade standards¹⁹.

Point-by-point response to reviewer 2:

1. I believe there is a typo in Figure 5's legend. It states that an N of 3 was used. An N of 3 would be ok for histology, but an N of 8 or more would be needed for the lower urinary tract cytometry. Additionally, are you sure those error bars are standard deviation? Those standard deviations are minimal compared to others published for cytometry (Kuo 2001, Urologia Internationalis). Please clearly state how many animals were used in the functional assessments in the Methods section and the figure legend.

Response: Thank you for your kind reminder and correction. We increased the number of cytometry experiments to 10 per group, and the error bars represent standard deviation (SD), and after increasing the number, we found that there were some differences within the groups due to the individual differences of different rats. After increasing the number, it was found that there were some differences within the groups due to the individual differences of different rats, and the error bars were increased. In addition, the number of animals used for each functional experiment has been clearly stated in the Methods section and in the figure legends.

Figure 5. (C) Histograms demonstrating LPP values for different groups at different time points. Data are expressed as the mean \pm standard deviation (SD) (n = 10). ns, not significant (p > 0.05), *p < 0.05, ***p < 0.001, ****p < 0.0001.

2. More details are needed in the methods section to help readers understand the experiments. Please provide more information about the vaginal distension (VD) and ovariectomy. For the VD, please give the size of the dilators used, the balloon size, the volume it was filled with, and how long it was inflated. These can vary from lab to lab, so it is best to state them clearly. Please justify starting the experiment with 4-week-old rats. Most studies use older rats. Was a separate cohort of rats used for 4 weeks vs 8 weeks? What was the flow rate for the cytometry? How many cycles did you run for data?

Response: Thank you for your suggestions to help refine the manuscript. Briefly, rats were anaesthetized by exposure to 20 % ketamine, and both right and left ovaries were then removed. A 12F catheter (Bard, Kulim, Malaysia) was inserted into the vagina, followed by injection of 2.5 mL of PBS into the air sac to compress the vaginal wall. The F12 catheter was leveled and attached with 120 mg of weight for 4 h before removal. The rats were fed normally for 4 weeks, after which an urodynamics examination was performed.

Experimental design: In this study, we chose to use 4-week-old female rats to construct SUI models, mainly due to the fact that 4-week-old rats are more physiologically and metabolically stable, with less inter-individual variability, and more vigorous development and regeneration of the neuromuscular system. In addition, although the 4W rats were relatively young at the time of modeling, we evaluated stress urinary incontinence at 4W after modeling (i.e., 8-week-old rats) due to the pelvic floor musculature and nerve damage, as well as the degradation of urethral sphincter

structure/function due to the decline in estrogen levels that takes some time. At this time, the rats gradually entered adulthood, and the development of organs and functions was stabilized, and the timing of the composite systemic injection treatment was selected at this time point.

Cytometry experiments: the pump flow rate was 6 ml/h. Twenty cycles were run and three stable cycles were selected for graphing.

(Page 32, line 33-39) All experiments were conducted using 4-week-old female Sprague – Dawley (SD) rats. The use of female animals was based on the clinical relevance of SUI, which predominantly affects women. SD rats were divided into 5 groups: SUI, PC, PCdL, PCdL/ADSCs (PEG) and PCdL/ADSCs (Mg) groups (three rats per group), and the rat SUI model was constructed by vaginal balloon dilatation and removal of bilateral ovaries. Briefly, rats were anaesthetized by exposure to 20 % ketamine, and both right and left ovaries were then removed. A 12F catheter was inserted into the vagina, followed by injection of 2.5 mL of PBS into the air sac to compress the vaginal wall. The F12 catheter was leveled and attached with 120 mg of weight for 4 h before removal.

(Page 33, line 7-9) The pump flow rate was 6 ml/h. Twenty cycles were run and three stable cycles were selected for graphing. The pressure at which leakage occurs was defined as the LPP (n=10).

3. Please provide more details about the stem cells. How many times did you expand them? How many animals were used for harvesting? What were the animals' ages and sexes? Did you select for stem cells, and if so, what markers did you use? How many separate batches of cells were harvested?

Response: We appreciate your thorough and valuable feedback aimed at improving the quality of our article. Expansion times: After initial adherence at passage 0 (P0), cells were cultured with medium changes every 2–3 days and passaged when confluence reached ~80%. Expansion was terminated at passage 4 (P4), which was used for subsequent experiments. As previously reported, rADSCs at P4 retain a stable morphology and demonstrate robust proliferative capacity and multilineage differentiation potential²².

Number of batches: To minimize inter-animal and inter-batch variability, three independent batches of ADSCs were isolated and expanded, each from three individual female Sprague–Dawley (SD) rats (n=9 in total). Prior to MOF programming and injection therapy, pooled ADSCs from each batch were thoroughly mixed and then equally distributed across experimental conditions. This approach helps ensure uniformity in cellular activity and biofunction across all treated groups.

Animal strain, sex and age: Female SD rats were selected based on anatomical similarity to the human pelvic floor, and 4-week-old animals were chosen due to their superior adipose stem cell

proliferation and differentiation capacity, which is particularly advantageous for downstream programming and therapeutic delivery.

For primary isolation, enzymatically digested adipose tissue was processed to obtain a single-cell suspension. Following 24 hours of incubation, non-adherent cells were removed, and adherent populations were retained, serving as an initial enrichment step for ADSCs²³.

According to the foundation of the previous research of this group^{24,25}, to validate stem cell identity, flow cytometry analysis revealed that the isolated cells expressed high levels of mesenchymal stem cell markers CD90, CD105, and CD44, while hematopoietic markers CD34 and CD45 were absent (Fig. F). Moreover, trilineage differentiation assays confirmed their adipogenic, osteogenic, and chondrogenic potential. Oil Red O staining revealed lipid droplet accumulation (Fig. C), indicating adipogenesis; Alizarin Red staining confirmed calcium deposition during osteogenic differentiation (Fig. D); and Alcian Blue staining demonstrated glycosaminoglycan production during chondrogenesis (Fig. E), altogether validating the multipotency of the isolated rADSCs.

Proliferation and differentiation of ADSCs. (A) Bright-field images of ADSCs cultured for 1 day. (B) Bright-field images of ADSCs cultured for 5 days. (C) Adipogenic differentiation of ADSCs. (D) Osteogenic differentiation of ADSCs. (E) Chondrogenic differentiation of ADSCs. (F) Flow cytometry analysis of surface-marker expression on ADSCs. Scale bars: 100 μ m.

(Page 30, line 24-33) Briefly, ADSCs were extracted from SD rats (female, 4-week-old) fresh adipose tissue in low sugar DMEM medium containing 15% FBS, 1% penicillin/streptomycin and 20 mg/ml vitamin C for continuous culture. To minimize inter-animal and inter-batch variability, three independent batches of ADSCs were isolated and expanded, each from three individual female Sprague–Dawley (SD) rats (n = 9 in total). Prior to MOF programming and injection therapy, pooled ADSCs from each batch were thoroughly mixed and then equally distributed across experimental conditions. The obtained ADSCs (P4) membrane sheets were immersed in decellularization solution for decellularization for 72 h.

4. Please provide a table with all the abbreviations and their meaning.

Response: We appreciate your suggestion to enhance the understanding of the manuscript.

Abbreviation	Full name	Abbreviation	Full name
SUI	Stress urinary incontinence	ADSCs	Adipose-derived stem cells
ECM	Extracellular matrix	NMJ	Neuromuscular junction
t-BA	tert-butyl acrylate	NIPAm	N-isopropylacrylamide
PNIPAm-C	Carboxyl-modified poly-NIPAm	dECM	decellularized ECM
EDC	1-ethyl-3-[3-dimethylaminopropyl] carbodiimide hydrochloride	NHS	N-hydroxysuccinimide
PCdL	PNIPAm-C/leucine/dECM	ZIF-8	Zeolitic imidazolate framework-8
MOFs	Metal-organic frameworks	Mg ²⁺	Magnesium ions
NPs	Nanoparticles	LPP	Leak point pressure
qRT-PCR	Quantitative reverse transcription polymerase chain reaction	WB	Western blot
Leu	Leucine	PBS	Phosphate Buffered Saline
2-MIM	2-methylimidazole	PEG	Polyethylene glycol
MgCl ₂	Magnesium chloride	RNA	Ribonucleic acid
a.u.	Arbitrary unit	DEGs	Differentially expressed genes
CCK8	Cell Counting Kit 8	KEGG	Kyoto Encyclopedia of Genes and Genomes
GO	Gene Ontology	HUVECs	Human umbilical vein endothelial cells
VEGF	Vascular endothelial growth factor	PDGF-BB	Platelet-derived growth factor-BB
EGF	Epidermal growth factor	HGF	Hepatocyte growth factor
PC	PNIPAm-C polymer	PC/Leu	PNIPAm-C/leucine polymer
PC/Leu/dECM	PNIPAm-C/leucine/ADSCs sheets-derived dECM powder polymer	PI	Propidium Iodide
3D	Three dimensions	SD	Standard deviation

HE	Hematoxylin-eosin staining	α -SMA	α -smooth muscle actin
TUBB3	tubulin β 3	α -BTX	α -bungarotoxin
PCdL/ADSCs (PEG)	PCdL+ADSCs+ZIF- 8/PEG200	PCdL/ADSCs (Mg)	PCdL+ADSCs+ZIF- 8/PEG200@Mg
CM	Congenital myasthenia	MUSK	Skeletal receptor tyrosine- protein kinase
DOK7	Docking Protein 7		

5. For all figures, please ensure you define all abbreviations in the legend. For Figure 2 in panel b, I am assuming a.u. stands for arbitrary unit. Additionally, in the corners of panel c immunofluorescent images, there is an indicator that they should be more prominent in size because they are currently unreadable. If they are not required, remove them.

Response: Thanks for your valuable questions.

Figure 2 in panel b: a.u. stands for arbitrary unit.

Figure 2 in panel c: Since the scale in Fig. 2, panel c comes with the machine, it cannot be adjusted for sharpness and size. To make it easier for the reader to understand, we have redrawn a clearer scale in the lower right corner of each figure.

6. For Figure 3, were replicates for the n of 3? Were three separate batches of cells used, or were they from all the identical batches?

Response: N=3 means three replicates per group. We expanded ADSCs from the same batch into nine dishes of P4 cells. Three of the dishes were left untreated, three were phagocytosed with ZIF-8/PEG200 NPs, three were phagocytosed with ZIF-8/PEG200@Mg NPs, and finally the cells were collected to obtain three groups of three cell clusters each for RNA sequencing. qRT-PCR tests were performed for the validation of several key genes identified by sequencing, again using cells from the same batch of ADSCs, divided into three groups of three replicates each.

7. In Figure 4, panel F, no Propidium Iodide (PI) stained cells are visible in the image, but your pie charts state that some PI-positive cells should be present. I recommend showing an example image with some PI-positive cells to demonstrate that the PI staining was done correctly.

Response: Thank you for your suggestions to help refine the manuscript. Due to the low percentage of dead cells and the dark red fluorescent staining, no cells with significant PI positivity were seen in the images. We added a PI-positive photo to the Supplementary Material to demonstrate that the

PI staining was done correctly.

Figure S12. PI-positive photo to demonstrate that the PI staining was done correctly.

8. In Figure 5 panel B, please include a time scale for each of the bladder traces or make all the traces to the same time scale and provide that scale in one of the traces.

Response: Thanks for your valuable suggestions. We have added a time scale for each of the bladder traces.

9. In Figure 8 panel A, the diagram is too small to read. I recommended either making a more simplified version or removing it.

Response: We have removed Figure 8 panel A and presented the figure in the supplemental materials.

10. Please consider doing some triple staining for NMJs and nerve axons. Many groups do stain to determine the number and percentage of innervated NMJs.

Response: We appreciate your thorough feedback aimed at enhancing the quality of our article. We have added triple staining for NMJs and nerve axons to facilitate the reader's understanding of the effects of the injectable system in promoting localized nerve repair and recovery of NMJ function.

Figure S18. Representative pictures of NMJ identified by α -Bungarotoxin (α -BTX, postsynaptic) and SV2 (presynaptic).

Figure S19. NMJ occupancy of local tissues after 8 weeks of treatment with different injection systems. Data are expressed as the mean \pm SD ($n = 3$). ns, not significant ($p > 0.05$), ** $p < 0.01$, *** $p < 0.001$.

(Page 20, line 23-27) Moreover, analysis of NMJ occupancy serves as a quantitative indicator of neuromuscular junction innervation and offers sensitive resolution for evaluating the extent of nerve regeneration. In this study, triple immunostaining for NMJs and axons revealed a markedly higher occupancy rate in the PCdL/ADSCs (Mg) group at 8 weeks post-injection, indicating enhanced NMJ maturation and neuroregenerative outcomes (Figure S18, S19).

Point-by-point response to reviewer 3:

1. The abstract is overly verbose and does not clearly emphasize the novelty of the work. The purpose, main findings, and significance of the study are not presented with sufficient clarity.

(Page 2, line 2-16) Stress urinary incontinence (SUI) remains a significant clinical challenge due to the lack of strategies that simultaneously address muscle degeneration, neurogenic atrophy, and vascular deficits. Here, we report an innovative injectable system that combines a thermo-responsive poly(N-isopropylacrylamide)-COOH/leucine/decellularized extracellular matrix hydrogel with adipose-derived stem cells (ADSCs) pre-programmed by zeolitic imidazolate framework-8/polyethylene glycol 200@magnesium (ZIF-8/PEG200@Mg) nanoparticles. In vitro, programmed ADSCs exhibited enhanced neurogenic differentiation, while the hydrogel matrix supported robust myogenic activity and cell viability. In vivo, injection of the composite system into a rat SUI model significantly improved leak point pressure (LPP) and restored urethral sphincter function. Mechanistic analyses revealed upregulation of muscle regeneration (e.g., Myoz1, Smyd1) and neurogenesis/neuromuscular junction stabilization (NMJ) stabilization genes (e.g., Ky, Dok7, Musk), highlighting a

coordinated multi-lineage regenerative process. This work establishes, for the first time, an integrated “regeneration-plus-support” injectable strategy, offering a regenerative medicine-based approach that surpasses conventional bulking or sling therapies for SUI.

2. The manuscript suffers from inconsistent transitions between sections. For instance, the introduction of MOF nanoparticles mixes background information and experimental details, hindering coherence.

Response: We appreciate your suggestion to enhance the understanding of the manuscript. We've revised the introductory section to improve the flow of the chapters and increase the coherence of the content.

(Page 4-5, line 15-39, 1-14) To address the outstanding challenges and fill existing gaps in the field, we herein report a “regeneration plus mechanical support” injectable strategy for the treatment of SUI for the first time. Specifically, we engineered a thermoresponsive hydrogel-based delivery system incorporating programmed stem cells, which not only offers localized structural support to the atrophic urethral sphincter but also simultaneously promotes myogenesis, angiogenesis, and neurogenesis—targeting the core pathological hallmark of sphincter degeneration to enable functional reconstruction and recurrence prevention. From a materials design perspective, we synthesized carboxyl-modified PNIPAm (PNIPAm-C) hydrogel by copolymerizing N-isopropylacrylamide (NIPAm) with tert-butyl acrylate (t-BA), followed by trifluoroacetic acid-mediated deprotection. Leucine, a known activator of the mTOR pathway with anabolic effects on muscle protein synthesis and inhibitory effects on proteolysis, was covalently grafted onto PNIPAm-C via EDC/NHS crosslinking to endow the hydrogel with intrinsic myogenic bioactivity^{26,27}. To further enrich the local biochemical microenvironment, decellularized ECM (dECM) powders derived from ADSCs sheets were integrated into the hydrogel network. The resulting thermosensitive injectable system employed both hydrophobic associations and amide bond-mediated dual crosslinking to form a mechanically robust and biologically active matrix. Recognizing the limited neuroregenerative capacity of conventional stem cell therapies, metal-organic framework (MOF) nanoparticles were introduced to pre-program stem cells. We chose ZIF-8, which has excellent biocompatibility and stability^{28,29}, as the base material and loaded magnesium ions (Mg^{2+}) to enhance its neural repair and angiogenesis functions^{30,31}. Considering the unstable release problem that might be triggered by the direct incorporation of MOF particles into the hydrogel, we pre-delivered ZIF-8/PEG200@Mg nanoparticles into the interior of ADSCs cells to achieve stem cell pro-vascularization and pro-neurogenesis functional programming before co-injecting them with the hydrogel, which ensured the sustained and stable biological effects.

Overall, the treatment of SUI remains a complex and multifactorial challenge. In a rat model of SUI,

localized injection of the engineered injectable composite system into the periurethral region at the bladder neck significantly restored urethral sphincter function, as evidenced by improved LPP at both 4- and 8-weeks post-treatment. Histological analyses further demonstrated robust regeneration of both striated and smooth muscle fibers, accompanied by enhanced vascularization, nerve fiber restoration, and reformation of NMJs. RNA-sequencing, qRT-PCR, and western blot (WB) analysis revealed that leucine and dECM powder effectively upregulated myogenic markers such as Myozi1 and Smyd1, while preprogrammed ADSCs significantly enhanced expression of neuroregenerative and NMJ-stabilizing genes, including Ache, Dok7, and Musk. Collectively, this study introduces a paradigm-shifting “regeneration plus mechanical support” strategy that outperforms conventional approaches based solely on volumetric bulking or static structural reinforcement (Scheme 1). By simultaneously promoting muscle reconstruction, angiogenesis, and neural repair of the urethral sphincter, our work lays the groundwork for a promising regenerative medicine-based therapy with translational potential for SUI patients.

3. Despite the manuscript's innovative use of MOF nanoparticles and ADSCs hydrogel systems, this synergy is not emphasized enough. The rationale behind the combination is not clearly articulated, particularly regarding how it addresses the limitations of existing SUI treatments.

4. The descriptions of the PNIPAm hydrogel and MOF nanoparticles are too general. The linkage between their specific properties (e.g., thermosensitivity, crosslink density, magnesium delivery) and their functional advantages in nerve and muscle regeneration is insufficiently discussed.

Response to question 3 and 4: Thanks for your valuable suggestions. We have added a description of the synergistic interaction of MOF nanoparticles with ADSCs hydrogel systems and a description of PNIPAm hydrogels, exploring the functional advantages conferred by the specific properties of the injection system in nerve and muscle regeneration

(Page 27-28, line 12-39, 1-25) To address the longstanding challenge of simultaneously promoting muscle regeneration and neural repair in the treatment of SUI, we report a pioneering injectable regenerative platform integrating MOF-programmed stem cells with a thermoresponsive hydrogel matrix. This system not only provides immediate mechanical support to the compromised urethral sphincter but also delivers endogenous chemical and biological cues that coordinately activate myogenic, angiogenic, and neurogenic repair pathways. By targeting the multifaceted pathophysiology of SUI, this strategy aims to achieve more complete functional restoration of the sphincter and reduce the risk of recurrence. The thermoresponsive PNIPAm-COOH hydrogel used in this study exhibits precise sol–gel transition behavior. Upon reaching physiological temperature

(lower Critical Solution Temperature $\approx 32^\circ\text{C}$), the PNIPAm backbone undergoes a sharp phase separation, enabling the injectable precursor solution to rapidly convert into a stable three-dimensional network in situ after administration. This facilitates localized shaping and mechanical reinforcement of the urethral sphincter region. To enhance mechanical integrity, leucine was covalently grafted onto the hydrogel via EDC/NHS-mediated amide bond formation, thereby increasing the crosslinking density of the matrix. This dual modification preserved the thermal responsiveness while significantly improving tensile and compressive moduli, providing sufficient stiffness to withstand repeated increases in abdominal pressure and better support sphincter regeneration. Notably, leucine was stably conjugated to the PNIPAm-C backbone rather than simply blended, leveraging its intrinsic ability to activate the mTOR signaling pathway. As a result, the hydrogel system markedly upregulated myogenic regulatory genes such as Myo1 and Smyd1, enhancing myogenic differentiation and simultaneously suppressing protein catabolism—demonstrating superior regulatory efficiency over physical mixtures. Previous studies have demonstrated that ADSCs, enriched with a variety of trophic factors, can accelerate muscle regeneration and restore urethral sphincter function. However, their capacity to promote neuronal repair and stabilize NMJs remains limited. To address this shortcoming, we innovatively employed ZIF-8/PEG200@Mg MOF NPs for pre-programming of ADSCs. ZIF-8 nanoparticles, known for their high biocompatibility, structural stability, and tunable surface properties, serve as an ideal nano-carrier platform³². Mg^{2+} , as essential physiological cofactors, play crucial roles in synaptic signal transmission, NMJ stabilization, axonal outgrowth, and endothelial cell-mediated angiogenesis. To leverage these effects, we encapsulated Mg^{2+} within the ZIF-8 framework and modulated PEG200 surface chemistry and porosity to achieve sustained ion release. This strategy maintains a local Mg^{2+} concentration within an optimal therapeutic window for both neuroregeneration and vasculature remodeling. By allowing ADSCs to internalize ZIF-8/PEG200@Mg nanoparticles prior to hydrogel encapsulation, we endowed the cells with a dual programming effect—pro-neurogenic and pro-angiogenic—while avoiding the risks associated with uncontrolled burst release of free nanoparticles. This cell-internalized delivery approach significantly enhanced the therapeutic efficacy of MOF-based nanomaterials. Synergistic effects were validated in both in vitro and in vivo models. ADSCs programmed with ZIF-8/PEG200@Mg showed upregulation of neurogenic markers such as Dok7 and Musk. In a co-culture model with HUVECs, these ADSCs also promoted robust expression of angiogenic genes. In vivo, SUI rats treated with the composite injectable exhibited substantial improvements in LPP at both 4- and 8-weeks post-injection, accompanied by restored voiding function. Histological and immunofluorescent analyses revealed extensive regeneration of both striated and smooth muscle

fibers, augmented neovascularization, and reestablishment of NMJ architecture. Intriguingly, transcriptomic and protein-level analyses (RNA-seq, qRT-PCR, and WB) confirmed activation of NMJ-related pathways. As the functional interface between motor neurons and skeletal muscle, NMJs are essential for neuromuscular signal transmission and maintenance of muscle tone. Their degeneration can lead to denervation and atrophy^{33,34}. Remarkably, ADSCs programmed with Mg²⁺-releasing MOFs significantly upregulated key NMJ stabilization genes such as *Ky*, *Dok7*, and *Musk*. This is particularly notable, mutations in *Dok7* may cause abnormal phosphorylation and activation of MUSK proteins, leading to NMJ dysfunction thereby causing muscle weakness and muscle atrophy^{35,36}. Upregulation of *Dok7* and *Musk* thus provides a robust molecular basis for restoring sphincter function via NMJ stabilization and regeneration.

5. Figures, especially those involving RNA sequencing data (e.g., Figures 7 and 8), include dense information without sufficient explanation or cross-referencing in the text.

Response: We appreciate your suggestion to enhance the quality of the manuscript. We have interpreted the data graph section of the article, adding textual explanations and cross-references to improve readability of the article.

(Page 22, line 1-39) To further elucidate the molecular mechanisms by which the injectable system promotes urethral sphincter regeneration and neurogenesis, we conducted RNA-seq on urethral tissues harvested seven days post-injection. DEG analysis was performed to decode biological processes involved in tissue repair. A heatmap of DEGs (Figure 7A; Figure S21A) visually highlighted widespread transcriptional alterations across treatment groups versus the SUI control, with upregulated genes marked in red and downregulated in blue³⁷. Principal component analysis (PCA) revealed distinct transcriptomic separation among PCdL-, PCdL/ADSC(PEG)-, and PCdL/ADSC(Mg)-treated groups (Figure 7B–D; Figure S20A; Figure S21B), indicating that each formulation significantly modulated the gene expression landscape³⁸. Complementary volcano plots (Figure 7B–D; Figure S21C) mapped log₂FC against –log₁₀P-values, further underscoring the intensity and directionality of transcriptional responses induced by each injectable strategy³⁹. Notably, DEG quantification (Figure S20B) demonstrated the PCdL/ADSC(Mg) group harbored the highest number of DEGs, suggesting a more robust and comprehensive reprogramming effect on urethral tissues⁴⁰. To dissect the functional implications of upregulated DEGs, we conducted GO and KEGG pathway enrichment analyses (Figure 7E–G; Figure S21D). In the PCdL group, upregulated genes were predominantly enriched in immune response, muscle contraction, and wound healing, indicating that the hydrogel scaffold alone could elicit localized immunomodulation and tissue remodeling. GO enrichment under Cellular Component terms highlighted associations with striated muscle thin filament, extracellular space, and sarcomere, supporting structural

remodeling of contractile apparatuses (Figure 7E). Concurrently, KEGG analysis revealed activation of proliferative signaling cascades, such as the MAPK pathway, likely mediating hydrogel-driven tissue repair⁴¹. In the PCdL/ADSC(PEG) group, DEGs were further enriched in muscle cell contraction and muscle fiber development, reflecting the pro-myogenic influence of stem cell inclusion³⁷. Enrichment in muscle myosin complex and sarcomere-associated regions suggests reinforced muscle functionality. In terms of molecular function, FATZ binding and telethonin binding were significantly upregulated. FATZ proteins stabilize Z-disc integrity and mediate mechanical signaling⁴², while telethonin is crucial for sarcomere assembly and mechanical signaling transduction⁴³—together implying enhanced sarcomeric maturity and contractile readiness (Figure 7F). Of particular note, the PCdL/ADSC(Mg) system uniquely induced neuroregenerative signatures, including positive regulation of neuron projection development and neuromuscular junction development (Figure 7G). These neurogenic processes are vital for re-establishing neural circuits and restoring sphincter innervation, distinguishing this group mechanistically from the others⁴⁴. GO cellular component analysis revealed marked upregulation of neuromuscular junction, while KEGG pathway analysis highlighted enhanced activity in motor protein dynamics and the calcium signaling pathway (Figure S22)—both pivotal in synaptic transmission and neuromuscular coordination. Finally, gene set enrichment analysis (GSEA) corroborated that both PCdL/ADSC(PEG) and PCdL/ADSC(Mg) significantly upregulated key signatures associated with muscle contraction and muscle regeneration (Figure S23), providing robust transcriptional evidence for the multilayered regenerative capacity of the engineered hydrogel system.

(Page 24, line 1-38) To explore the unique neurogenic potential of the PCdL/ADSC(Mg) injectable system and assess the *in vivo* biological effects of ZIF-8/PEG200@Mg-mediated ADSC programming, we performed differential gene expression analysis between the PCdL/ADSC(Mg) and PCdL/ADSC(PEG) groups (Figure 8A). The volcano plot (Figure 8B) revealed 875 significantly upregulated genes, defined by a fold change ≥ 2 and $P < 0.05$, which were included in downstream enrichment analyses. GO enrichment in the *Biological Process* category (Figure 8C) highlighted strong enrichment in pathways related to neural transmission, including the acetylcholine receptor signaling pathway and chemical synaptic transmission, postsynaptic. These processes are pivotal for synaptic transmission and neuromuscular communication. In particular, acetylcholine signaling at the NMJ plays a central role in mediating muscle excitation and contraction⁴⁵. Postsynaptic chemical synaptic transmission represents the cascade through which neurotransmitters activate receptors, crucial for neuromuscular coordination and plasticity⁴⁶. Interestingly, enrichment of transition between fast and slow fiber suggests potential remodeling of

muscle fiber phenotypes, indicative of myofiber reprogramming under regenerative conditions⁴⁷. GO analysis in Cellular Component and Molecular Function categories also revealed significant upregulation of genes associated with acetylcholine-gated channel complex and cation-selective channel activity, implying enhanced functionality of postsynaptic ion channels and synaptic responsiveness⁴⁸. Further KEGG pathway analysis indicated significant upregulation of the Neuroactive ligand-receptor interaction pathway (Figure 8D), which encompasses a wide array of neurotransmitter ligands and receptors. This suggests that ZIF-8/PEG200@Mg programmed ADSCs may modulate ligand-receptor networks to amplify and fine-tune neural signaling⁴⁹. Taken together, these results suggest a mechanistic model in which the PCdL/ADSC(Mg) system activates calcium ion channels, thereby promoting intracellular calcium influx. This elevation in Ca²⁺ levels triggers synaptic vesicle fusion with the presynaptic membrane, facilitating the release of acetylcholine (ACh) into the synaptic cleft. The binding of ACh to postsynaptic nicotinic acetylcholine receptors (nAChRs) initiates sodium influx, generates a localized action potential, and induces contraction of the urethral sphincter to restore continence and muscle tone⁵⁰. To mechanistically validate the effects of this injectable system on myogenesis and neurogenesis, we performed qRT-PCR (Figure 8E) and WB (Figure 8F) to monitor the expression dynamics of key regulatory genes. For muscle contraction and regeneration, genes including *Atp2a1* (encoding SERCA1, responsible for sarcoplasmic Ca²⁺ reuptake), *Klhl41* (regulates protein degradation and muscle fiber homeostasis⁵¹), *Smyd1* and *Myoz1* (involved in muscle differentiation and remodeling), as well as *Myh1* and *Mef2c* (critical for myofiber lineage specification⁵²), were significantly upregulated. For neurogenesis and NMJ stabilization, we observed increased expression of *Musk* and *Dok7*, key structural regulators of NMJ assembly, as well as *Ky* (involved in calcium sensitivity and nerve-muscle interface), *Chrnd* and *Ache* (modulators of AChR clustering and degradation⁵²), and *Mapt* (a microtubule-associated protein essential for axonal stability and synaptic plasticity).

(Page 25-26, line 12-15, 1-15) Interestingly, recent studies have identified a strong association between mutations in *Dok7* and congenital myasthenic (CM) syndromes⁵³, a group of neuromuscular disorders characterized by impaired neuromuscular transmission. *Dok7* encodes an adaptor protein essential for the maintenance and function of NMJs⁵⁴. Loss-of-function mutations in *Dok7* result in defective NMJ formation, leading to early-onset muscle weakness and, in severe cases, perinatal lethality. Mechanistically, *Dok-7* functions as an adaptor protein that anchors to muscle, skeletal receptor tyrosine-protein kinase at its N-terminal domain to stabilize MUSK phosphorylation⁵⁵. As a receptor tyrosine kinase, MUSK serves as a key regulator of synaptic differentiation and orchestrates presynaptic specialization^{56,57}. Loss-of-function mutations in *Dok7* cause severe defects in MUSK phosphorylation and activation, thereby preventing the recruitment

of essential NMJ-associated proteins, such as adapter molecule CRK, and ultimately leading to impaired synaptic integrity and muscle weakness⁵⁸. In this study, we observed that the PCdL/ADSC(Mg) injectable system significantly upregulated both Dok7 and Musk expression in the urethral sphincter tissue, which likely contributed to enhanced NMJ stability and function. This molecular activation was corroborated at both the mRNA and protein levels, suggesting that the PCdL/ADSC(Mg) platform exerts a dual regenerative effect by promoting muscle regeneration and supporting neuromuscular reinnervation. These findings offer mechanistic insight into the therapeutic efficacy of this system in SUI repair and highlight its potential for restoring neuromuscular functionality through targeted molecular modulation.

6. Although statistical significance is provided for most experiments, some datasets lack detailed quantitative analysis. The functional evaluation of NMJ restoration, for example, would benefit from additional metrics like synaptic density or acetylcholine receptor characterization.

Response: We appreciate your thorough feedback aimed at enhancing the quality of our article. We have added triple staining for NMJs and nerve axons to facilitate the reader's understanding of the effects of the injectable system in promoting localized nerve repair and recovery of NMJ function.

Figure S18. Representative pictures of NMJ identified by α -Bungarotoxin (α -BTX, postsynaptic) and SV2 (presynaptic).

Figure S19. NMJ occupancy of local tissues after 8 weeks of treatment with different injection systems. Data are expressed as the mean \pm SD ($n = 3$). ns, not significant ($p > 0.05$), ** $p < 0.01$, *** $p < 0.001$.

(Page 20, line 23-27) Moreover, analysis of NMJ occupancy serves as a quantitative indicator of neuromuscular junction innervation and offers sensitive resolution for evaluating the extent of nerve regeneration. In this study, triple immunostaining for NMJs and axons revealed a markedly higher occupancy rate in the PCdL/ADSCs (Mg) group at 8 weeks post-injection, indicating enhanced NMJ maturation and neuroregenerative outcomes (Figure S18, S19).

7. The discussion does not delve deeply into the practical challenges of clinical translation, such as potential immune responses, large-scale production, long-term material stability, or variability across patient-specific applications of the treatment system.

Response: Thanks for your valuable suggestions. We have increased the readability of the article by adding an exploration of the clinical translation section of the article.

(Page 29, line 11-29) In addition, although the hydrogel materials demonstrated favorable biocompatibility and minimal cytotoxicity in both in vitro and in vivo animal models, their potential to elicit long-term immune responses upon implantation in humans remains to be rigorously assessed. While the decellularization process effectively removes immunogenic components such as residual DNA from the dECM, comprehensive immunological profiling is still necessary to ensure clinical safety and minimize potential adverse responses. Notably, the long-term in vivo stability of the hydrogel plays a pivotal role in determining its therapeutic durability. Standard in vitro degradation assays may not faithfully replicate the complex in vivo degradation kinetics. Therefore, longitudinal preclinical studies—extending over 6 to 12 months in large animal models—will be essential to monitor urethral function, local tissue remodeling, and sustained muscle and neural regeneration, as well as to evaluate chronic toxicity and mechanical persistence. These data will provide critical insights into the translational potential and in vivo durability of the injectable platform. Lastly, SUI is a multifactorial condition with substantial inter-patient heterogeneity, including variations in urethral mobility, sphincter function, and age-related tissue degeneration. These differences can profoundly affect the therapeutic response to biomaterial-based

interventions. Hence, for successful clinical translation, future therapeutic strategies employing this injectable system should be tailored to the patient's underlying etiology and physiological characteristics to optimize efficacy and improve patient satisfaction.

2. Forcales, S.V. Potential of adipose-derived stem cells in muscular Degenerative therapies. *Frontiers in aging neuroscience* **7**, 123 (2015).
3. Mizuno, H. The potential for treatment of skeletal muscle disorders With adipose-derived stem cells. *Current stem cell research & therapy* **5**, 133-136 (2010).
4. Ehu, Y., *et al.* Adipose-derived stem cell: A better stem cell than BMSC. *Cell Research* **18**, S165-S165 (2008).
5. Kesireddy, V. Evaluation of adipose-derived stem cells for tissue-Engineered muscle repair construct-mediated repair of a murine model of volumetric muscle loss injury. *International journal of Nanomedicine* **11**, 1461-1473 (2016).
6. Bahu, K.P., *et al.* Nanomaterials via ZIF-8: Preparations, catalytic and drug delivery applications. *Chemical Engineering Journal* **508**, 160663 (2025).
7. Chen, Y., *et al.* Injectable Nanocomposite Hydrogel for Accelerating Diabetic Wound Healing Through Inflammatory Microenvironment Regulation. *International journal of nanomedicine* **20**, 1679-1696 (2025).
8. Ansari, M.J., *et al.* Poly(N-isopropylacrylamide)-Based Hydrogels for Biomedical Applications: A Review of the State-of-the-Art. *Gels (Basel, Switzerland)* **8**(2022).
9. Watanabe, T., *et al.* Increased urethral resistance by periurethral injection of low serum cultured adipose-derived mesenchymal stromal cells in rats. *International journal of urology : official journal of the Japanese Urological Association* **18**, 659-666 (2011).
10. Capella, V., *et al.* Cytotoxicity and bioadhesive properties of poly-N-isopropylacrylamide hydrogel. *Heliyon* **5**, e01474 (2019).
11. Hao, AY. Stem Cell MicroRegenerative Medicine and Fibroblasts: Advances and perspectives. *Bioresource technology* **397**, 130502 (2024).
12. Lu, Y.T., *et al.* Engineering of Stable Cross-Linked Multilayers Based on Thermo-Responsive PNIPAM-Grafted-Chitosan/Heparin to Tailor Their Physiochemical Properties and Biocompatibility. *ACS applied materials & interfaces* **14**, 29550-29562 (2022).
13. Huang, J., Cheng, M., Ding, Y., Chen, L. & Hua, K. Modified vaginal dilation rat model for postpartum stress urinary incontinence. *The journal of obstetrics and gynaecology research* **39**, 256-263 (2013).
14. Kitta, T., *et al.* Effects of ovariectomy and estrogen replacement on the urethral continence reflex during sneezing in rats. *The Journal of urology* **186**, 1517-1523 (2011).
15. Hijaz, A., Daneshgari, F., Sievert, K.D. & Damaser, M.S. Animal models of female stress urinary incontinence. *The Journal of urology*

- 179**, 2103–2110 (2008).
16. Kovacevic, N., Lopes, N.N., Raffee, S. & Atiemo, H.O. Predicting Upper Urinary Tract Risk in the Neurogenic Bladder Patient. *Current Bladder Dysfunction Reports* **15**, 66–71 (2020).
 17. Capobianco, G., *et al.* Efficacy and effectiveness of bulking agents in the treatment of stress and mixed urinary incontinence: A systematic review and meta-analysis. *Maturitas* **133**, 13–31 (2020).
 18. Wasenda, E.J., Kirby, A.C., Lukacz, E.S. & Nager, C.W. The female continence mechanism measured by high resolution manometry: Urethral bulking versus midurethral sling. *Neurourology and urodynamics* **37**, 1809–1814 (2018).
 19. Tian, J., *et al.* Regulatory perspectives of combination products. *Bioactive materials* **10**, 492–503 (2022).
 20. Asfour, V., Doumouchsis, S., Ghoneim, G., Emery, S. & Agur, W. Non-mesh stress incontinence surgery review: Bulking agents, autologous fascial slings and colposuspension. *Continence* **13**, 101727 (2025).
 21. Zhang, S., *et al.* Collagen type I-loaded methacrylamide hyaluronic acid hydrogel microneedles alleviate stress urinary incontinence in mice: A novel treatment and prevention strategy. *Colloids and surfaces. B, Biointerfaces* **222**, 113085 (2023).
 22. Farag, A., *et al.* Exploring the Potential Effects of Cryopreservation on the Biological Characteristics and Cardiomyogenic Differentiation of Rat Adipose-Derived Mesenchymal Stem Cells. *International journal of molecular sciences* **25**(2024).
 23. Mazini, L., Ezzoubi, M. & Malka, G. Overview of current adipose-derived stem cell (ADSCs) processing involved in therapeutic advancements: flow chart and regulation updates before and after COVID-19. *Stem cell research & therapy* **12**, 1 (2021).
 24. Wang, Y., *et al.* Use of bioactive extracellular matrix fragments as a urethral bulking agent to treat stress urinary incontinence. *Acta biomaterialia* **117**, 156–166 (2020).
 25. Yang, M., *et al.* Urine-Microenvironment-Initiated Composite Hydrogel Patch Reconfiguration Propels Scarless Memory Repair and Reinvigoration of the Urethra. *Advanced materials (Deerfield Beach, Fla.)* **34**, e2109522 (2022).
 26. Ham, D.J., Caldow, M.K., Lynch, G.S. & Koopman, R. Leucine as a treatment for muscle wasting: a critical review. *Clinical nutrition (Edinburgh, Scotland)* **33**, 937–945 (2014).
 27. Dodd, K.M. & Tee, A.R. Leucine and mTORC1: a complex relationship. *American journal of physiology. Endocrinology and metabolism* **302**, E1329–1342 (2012).
 28. Wang, Y., *et al.* Biomimetic ZIF-8 Nanoparticles: A Novel Approach for Biomimetic Drug Delivery Systems. *International journal of nanomedicine* **19**, 5523–5544 (2024).

29. Li, Z., Shao, Y., Yang, Y. & Zan, J. Zeolitic imidazolate framework-8: a versatile nanoplatform for tissue regeneration. *Frontiers in bioengineering and biotechnology* **12**, 1386534 (2024).
30. He, Z., *et al.* Logic-Based Diagnostic and Therapeutic Nanoplatform with Infection and Inflammation Monitoring and Microenvironmental Regulation Accelerating Wound Repair. *ACS applied materials & interfaces* **14**, 39172–39187 (2022).
31. Bose, S., Fielding, G., Tarafder, S. & Bandyopadhyay, A. Understanding of dopant-induced osteogenesis and angiogenesis in calcium phosphate ceramics. *Trends in biotechnology* **31**, 594–605 (2013).
32. Mohan, B., *et al.* Advanced electrochemiluminescent approaches for contaminant detection in food matrices using metal-organic framework composites. *Food chemistry* **470**, 142625 (2024).
33. Sanes, J.R. & Lichtman, J.W. Induction, assembly, maturation and maintenance of a postsynaptic apparatus. *Nature reviews. Neuroscience* **2**, 791–805 (2001).

3

4

.

35. Hamuro, J., *et al.* Mutations causing DOK7 congenital myasthenia abolish functional motifs in Dok-7. *The Journal of biological chemistry* **283**, 5518–5524 (2008).
36. Hallock, P.T., *et al.* Dok-7 regulates neuromuscular synapse formation by recruiting Crk and Crk-L. *Genes & development* **24**, 2451–2461 (2010).
37. Zhang, H., Huang, J., Liu, J., Li, Y. & Gao, Y. BMSC-sEV-derived miR-328a-3p promotes ECM remodeling of damaged urethral sphincters via the Sirt7/TGF β signaling pathway. *Stem cell research & therapy* **11**, 286 (2020).
38. Pokhilko, A., Nash, A. & Cader, Z. Common transcriptional signatures of neuropathic pain. *PAIN* **161**, 1 (2020).
39. Sun, W., *et al.* RNA sequencing profiles reveals progressively reduced spermatogenesis with progression in adult cryptorchidism. *Frontiers in endocrinology* **14**, 1271724 (2023).
40. Jiang, M., *et al.* Bone marrow stem cells secretome accelerates simulated birth trauma-induced stress urinary incontinence recovery in rats. *Aging* **13**, 10517–10534 (2021).
41. Huang, S., *et al.* Analysis of Genomic Alternative Splicing Patterns in Rat under Heat Stress Based on RNA-Seq Data. *Genes* **13**(2022).
42. Sponga, A., *et al.* Order from disorder in the sarcomere: FATZ forms a fuzzy but tight complex and phase-separated condensates with α -actinin. *Science advances* **7**(2021).
43. Jiang, H., *et al.* Functional analysis of a gene-edited mouse model

r

,

H

.

D

to gain insights into the disease mechanisms of a titin missense variant. *Basic research in cardiology* **116**, 14 (2021).

4

4

.

45. Soffi, E., *et al.* B cell-derived acetylcholine mitigates skin inflammation through $\alpha 7$ nAChR. *Neuron* **107**, 1051–1063 (2021).
46. Redhaud, S.G. *et al.* Double protein loss of the *titin* gene in mice leads to a severe myopathy. *Journal of Molecular Biology* **383**, 111–122 (2014).
47. Plotkin, D.L., Roberts, M.D., Haun, C.T. & Schoenfeld, B.J. Muscle Fiber Type Transitions with Exercise Training: Shifting Perspectives. *Sports (Basel, Switzerland)* **9** (2021).
48. Wu, Z.S., Cheng, H., Jiang, Y., Melcher, K. & Xu, H.E. Ion channels gated by acetylcholine and serotonin: structures, biology, and drug discovery. *Acta pharmacologica Sinica* **36**, 895–907 (2015).
49. Feng, Y., *et al.* Changes in goose hypothalamus under different photoperiods: RNA-Seq reveals new pathways and molecules. *Poultry science* **104**, 104883 (2025).
50. Yilmaz, A., *et al.* MuSK is a BMP co-receptor that shapes BMP responses and calcium signaling in muscle cells. *Science signaling* **9**, ra87 (2016).
51. Ehrlich, K.C., Baribault, C. & Ehrlich, M. Epigenetics of Muscle- and Brain-Specific Expression of KLHL Family Genes. *International journal of molecular sciences* **21** (2020).
52. Miretti, S., Volpe, M.G., Martignani, E., Accornero, P. & Baratta, M. Temporal correlation between differentiation factor expression and microRNAs in Holstein bovine skeletal muscle. *Animal : an international journal of animal bioscience* **11**, 227–235 (2017).
53. Oury, J., *et al.* Mechanism of disease and therapeutic rescue of Dok7 congenital myasthenia. *Nature* **595**, 404–408 (2021).
54. Okada, K., *et al.* The muscle protein Dok-7 is essential for neuromuscular synaptogenesis. *Science (New York, N. Y.)* **312**, 1802–1805 (2006).
55. Beeson, D., *et al.* Dok-7 mutations underlie a neuromuscular junction synaptopathy. *Science (New York, N. Y.)* **313**, 1975–1978 (2006).
56. Burden, S.J., Yumoto, N. & Zhang, W. The role of MuSK in synapse formation and neuromuscular disease. *Cold Spring Harbor perspectives in biology* **5**, a009167 (2013).
57. DeChiara, T.M., *et al.* The receptor tyrosine kinase MuSK is required for neuromuscular junction formation in vivo. *Cell* **85**, 501–512 (1996).

(1996).

58. Engel, A.G., Shen, X.M., Selcen, D. & Sine, S.M. Congenital myasthenic syndromes: pathogenesis, diagnosis, and treatment. *The Lancet. Neurology* **14**, 461 (2015).